# Disentangling Knowledge Representations for Large Language Model Editing

**Mengqi Zhang[1,2]\*, Zisheng Zhou[1]\*, Xiaotian Ye[3], Qiang Liu[4],**
**Zhaochun Ren[5], Zhumin Chen[1], Pengjie Ren[1†]**
[1]Shandong University
[2]Suzhou Research Institute of Shandong University
[3]School of Computer Science, Beijing University of Posts and Telecommunications
[4]New Laboratory of Pattern Recognition (NLPR)
 State Key Laboratory of Multimodal Artificial Intelligence Systems (MAIS)
 Institute of Automation, Chinese Academy of Sciences
[5]Leiden University
{mengqi.zhang, chenzhumin, renpengjie}@sdu.edu.cn
zisheng.zhou@mail.sdu.edu.cn,yexiaotian@bupt.edu.cn
qiang.liu@nlpr.ia.ac.cn,z.ren@liacs.leidenuniv.nl

## Abstract

Knowledge Editing has emerged as a promising solution for efficiently updating embedded knowledge in large language models (LLMs). While existing approaches demonstrate effectiveness in integrating new knowledge and preserving the original capabilities of LLMs, they fail to maintain fine-grained irrelevant knowledge, namely facts that share the same subject as edited knowledge but differ in relation and object. This challenge arises because subject representations inherently encode multiple attributes, causing the target and fine-grained irrelevant knowledge to become entangled in the representation space, and thus vulnerable to unintended alterations during editing. To address this, we propose DiKE, a novel approach that **Di**sentangles **K**nowledge representations for LLM **E**diting (DiKE). DiKE consists of two key components: a Knowledge Representation Disentanglement (KRD) module that decomposes the subject representation into target-knowledge-related and -unrelated components, and a Disentanglement-based Knowledge Edit (DKE) module that updates only the target-related component while explicitly preserving the unrelated one. We further derive a closed-form, rank-one parameter update based on matrix theory to enable efficient and minimally invasive edits. To rigorously evaluate fine-grained irrelevant knowledge preservation, we construct FINE-KED, a new benchmark comprising fine-grained irrelevant knowledge at different levels of relational similarity to the edited knowledge. Extensive experiments across multiple LLMs demonstrate that DiKE substantially improves fine-grained irrelevant knowledge preservation while maintaining competitive general editing performance.

## 1 Introduction

Large language models (LLMs) have garnered significant attention for their extensive knowledge storage and advanced reasoning capabilities (Zhao et al., 2024). However, the inherent noise in their training data and the continuous evolution of world knowledge often lead to inaccuracies and outdated information (Cao et al., 2021). To address these limitations, knowledge editing (Wang et al., 2024) has emerged as a promising solution, enabling precise and efficient updates to the knowledge embedded within LLMs. Among various approaches, parameter-modifying methods that directly update the internal parameters of LLMs are particularly appealing due to their ability to produce consistent outputs without requiring additional inference-time context or external memory.

---

\*Equal contribution.
†Corresponding author.

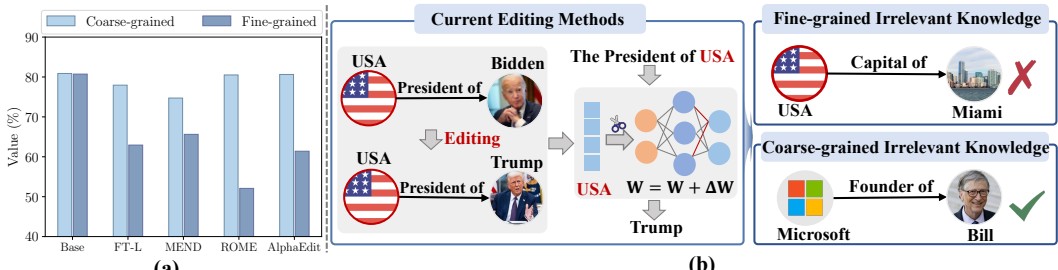

Figure 1: Knowledge editing can unintentionally affect fine-grained irrelevant knowledge. Figure (a): Preservation performance on fine-grained vs. coarse-grained irrelevant knowledge. Figure (b): Illustration of knowledge editing can unintentionally affect fine-grained irrelevant knowledge.

Representative methods include fine-tuning-based techniques (e.g., FT-L (Zhu et al., 2020)), meta-learning strategies (e.g., KE (Wang et al., 2025), MEND (Mitchell et al., 2022)), and locate-then-edit methods (e.g., ROME (Meng et al., 2022), MEMIT (Meng et al., 2023), AlphaEdit (Fang et al., 2025)). This work focuses specifically on parameter-modifying approaches.

A successful knowledge editing method should not only accurately inject the desired updates but also minimize unintended interference with existing irrelevant knowledge. Recent research (Geva et al., 2023) suggests that LLMs retrieve stored knowledge by recalling facts associated with specific subjects. These retrieval processes center around subject representations, which encapsulate extensive attribute information related to that subject. Consequently, irrelevant knowledge can be classified into two categories based on semantic proximity to the subject of edited knowledge: **fine-grained** and **coarse-grained** irrelevant knowledge. Fine-grained irrelevant knowledge comprises facts that share the same subject as the target knowledge but differ entirely in relation and object, rendering them logically independent of the knowledge being edited. For example, when updating the knowledge from *"The President of USA is Biden"* to *"The President of USA is Trump"*, a fine-grained irrelevant fact such as *"The capital of USA is Washington"* should remain unchanged despite sharing the subject *USA*. However, as they are encoded in the same subject representation, fine-grained irrelevant knowledge often becomes deeply entangled with the target knowledge within the model's parameter and representation spaces (Hernandez et al., 2024; Qin et al., 2024), rendering it particularly susceptible to unintended alterations during editing. In contrast, coarse-grained irrelevant knowledge refers to facts that involve completely different subjects, relations, and objects, such as *"Microsoft was founded by Bill Gates"*, and is typically distant enough from the target edit that inadvertent alteration is unlikely.

To conduct a preliminary analysis of how editing affects different types of irrelevant knowledge, we randomly sampled 1,000 edit instances. For each, we selected one fine-grained and one coarse-grained irrelevant fact and evaluated the performance of existing editing methods, FT-L (Zhu et al., 2020), MEND (Mitchell et al., 2022), ROME (Meng et al., 2022), and AlphaEdit (Fang et al., 2025), on LlaMA3(8B) in preserving them. As illustrated in Figure 1(a), the results show a marked discrepancy between the two categories, with fine-grained irrelevant knowledge being more vulnerable to unintended alterations. This observation highlights the inherent challenge of preserving fine-grained irrelevant knowledge during the editing process.

While current editing methods have shown promise in accurately injecting new knowledge and broadly preserving unrelated facts, **they frequently fail to adequately maintain fine-grained irrelevant knowledge.** As illustrated in Figure 1(b), current methods that directly manipulate the subject representation to perform knowledge edits tend to inadvertently degrade fine-grained irrelevant knowledge. Existing methods commonly attempt to mitigate this interference by imposing constraints derived from broadly sampled unrelated knowledge, such as text from Wikitext. However, these coarse-grained constraints are insufficiently precise and thus struggle to effectively prevent undesired alterations of fine-grained irrelevant knowledge.

To address the aforementioned challenge, we propose a novel locate-then-edit approach that **Di**sentangles **K**nowledge representations for LLM **E**diting (DiKE) to effectively preserve fine-grained knowledge unrelated to the target edit, as depicted in Figure 2. Specifically, we first introduce a Knowledge Representation Disentanglement (KRD) module (§3.1) that disentangles the subject representation into target-knowledge-related and target-knowledge-unrelated components

within the LLM's representation space. Subsequently, to inject target knowledge into LLMs without impacting fine-grained irrelevant knowledge, we develop a Disentanglement-based Knowledge Edit (DKE) (§3.2) module. This module performs editing operations on the target-knowledge-related representation while constraining the target-knowledge-unrelated representation to remain unchanged. Furthermore, leveraging matrix theory, we derive a rank-one update formula that satisfies the these constraints, enabling the efficient update of model parameters. To comprehensively evaluate the effectiveness of our method in preserving fine-grained irrelevant knowledge, we construct a new dataset, FINE-KED (§4.1), which categorizes test instances into three levels based on the relational semantic similarity between the edited knowledge and its fine-grained irrelevant counterparts. Extensive experiments using GPT2-XL (1.5B), GPT-J (6B) and LLaMA-3 (8B) on FINE-KED and COUNTERFACT demonstrate that DiKE effectively preserves fine-grained unrelated knowledge while achieving comparable general edit performance with other state-of-the-art editing methods. We summarize our contributions as follows:

- We categorize irrelevant knowledge into coarse-grained and fine-grained types, and identify the unique challenge of preserving fine-grained irrelevant knowledge in knoweldge editing.

- We propose DiKE, a novel knowledge editing method based on knowledge representation disentanglement, which enables precise editing of target knowledge while minimizing interference with fine-grained irrelevant knowledge. Furthermore, we derive a closed-form solution for parameter updates based on matrix theory, incorporating multiple constraints to ensure effective editing.

- We construct a new dataset FINE-KED, to thoroughly evaluate the ability of existing editing methods to preserve fine-grained irrelevant knowledge. Extensive experiments conducted on LLMs of varying sizes demonstrate the effectiveness of our proposed method.

## 2 PRELIMINARIES

### 2.1 AUTOREGRESSIVE LANGUAGE MODEL

We focus on autoregressive LLMs that generate text by predicting the next token sequentially. Let $F$ denote an LLM with $L$ transformer decoder layers, each consists of a multi-head attention (MHA) module and a feed-forward network (FFN). The hidden state representation at layer $l$ is computed as:

$$\mathbf{h}^l = \mathbf{h}^{l-1} + \mathbf{a}^l + \mathbf{v}^l, \tag{1}$$

where $\mathbf{a}^l$ and $\mathbf{v}^l$ denote the outputs of the MHA and FFN modules, respectively. The FFN output is given by:

$$\mathbf{v}^l = f(\mathbf{W}_{in}^l \cdot \mathbf{h}^{l-1}) \cdot \mathbf{W}_{out}^l, \tag{2}$$

where $\mathbf{W}_{in}^l$ and $\mathbf{W}_{out}^l$ are the parameter matrices of the first and second layers of the FFN, respectively, and $f(\cdot)$ is a non-linear activation function. For brevity, we omit the layer index $l$ in subsequent sections and denote $\mathbf{W} = \mathbf{W}_{out}^l$.

### 2.2 KNOWLEDGE EDITING VIA RANK-ONE UPDATES

**Knowledge Editing** aims to modify or inject single or multiple pieces of knowledge into LLMs without requiring full retraining. Each fact is typically represented as a triple $(s, r, o)$, where $s$ denotes subject, $r$ relation, and $o$ object, respectively. An edit sample is denoted as $e = (s, r, o, o^*)$, representing the update of the original knowledge $(s, r, o)$ to a new knowledge $(s, r, o^*)$.

**Rank-one Knowledge Editing**, exemplified by methods such as ROME and MEMIT (Meng et al., 2022; 2023), which follow a locate-then-edit paradigm. These methods assume that factual knowledge is encoded in the feed-forward networks (FFNs) of the model as key-value pairs, where the first FFN layer generates keys and the second layer uses these keys to produce values. To inject new knowledge, the FFN parameters $\mathbf{W}$ are updated to $\hat{\mathbf{W}}$, such that the model associates a new key-value pair $(\mathbf{k}_*, \mathbf{v}_*)$. To avoid corrupting unrelated knowledge, an auxiliary set of key–value constraints $\mathbf{K}_0 = [\mathbf{k}_1; \mathbf{k}_2; , \ldots, ; \mathbf{k}_p]$ and $\mathbf{V}_0 = [\mathbf{v}_1; \mathbf{v}_2; , \ldots, ; \mathbf{v}_p]$ is introduced, representing knowledge to be preserved.

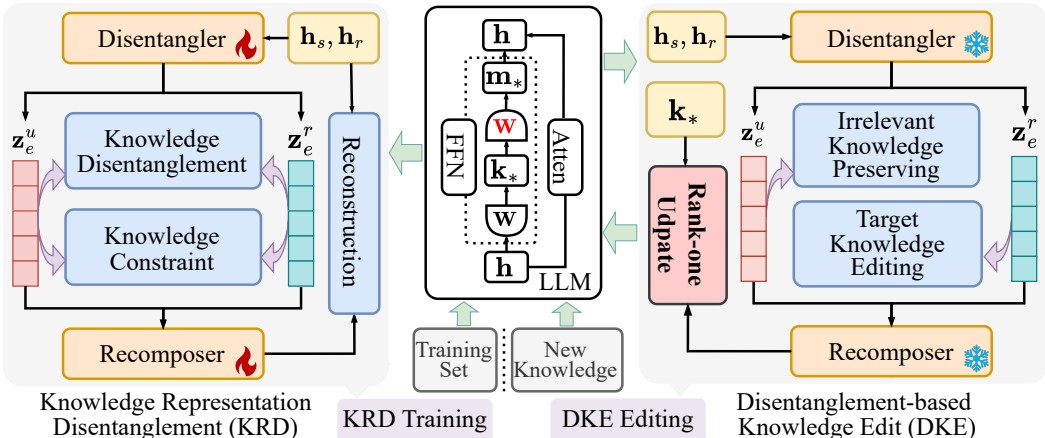

Figure 2: **Overview of the DiKE architecture.** The framework operates in two distinct phases. **(Left) KRD Training:** The module extracts subject and relation representations from the LLM and learns to disentangle them into target-knowledge-related and -unrelated components via optimizing disentanglement, constraint, and reconstruction objectives. **(Right) DKE Editing:** During the editing phase, the pre-trained Disentangler is frozen. The DKE module utilizes the disentangled representations to derive a closed-form rank-one parameter update (Eq. (19)), which injects new knowledge into the target-related component while explicitly constraining the unrelated component to preserve fine-grained irrelevant knowledge. **Note that the KRD module is pre-trained on the training set and does not require retraining for subsequent editing tasks.**

Adopting the least-squares formulation from MEMIT (Meng et al., 2023), the updated weights $\hat{\mathbf{W}}$ are obtained by solving the following objective:

$$\hat{\mathbf{W}} = \underset{\hat{\mathbf{W}}}{\mathrm{argmin}}(\|\hat{\mathbf{W}}\mathbf{k}_* - \mathbf{v}_*\|_F^2 + \|\hat{\mathbf{W}}\mathbf{K}_0 - \mathbf{V}_0\|_F^2). \tag{3}$$

The closed-form solution derived via the normal equation as in MEMIT, is given by:

$$\hat{\mathbf{W}} = \mathbf{W} + (\mathbf{v}_* - \mathbf{W}\mathbf{k}_*)\mathbf{k}_*^\top(\mathbf{K}_0\mathbf{K}_0^\top + \mathbf{k}_*\mathbf{k}_*^\top)^{-1}, \tag{4}$$

where $\mathbf{K}_0$ is estimated based on a sample of Wikipedia text Meng et al. (2023).

## 3 METHODOLOGY

In this section, we present our proposed method, DiKE, with its overall architecture illustrated in Figure 2. The DiKE framework comprises two core components: (1) *Knowledge Representation Disentanglement (KRD)*, which explicitly disentangles the subject representation into target-knowledge-related and -unrelated components within the LLM's representation space, and (2) *Disentanglement-based Knowledge Edit (DKE)*, which injects target knowledge into specific parameters using the disentangled representations while preserving the fine-grained irrelevant knowledge.

### 3.1 KNOWLEDGE REPRESENTATION DISENTANGLEMENT

Target edit knowledge and its fine-grained irrelevant knowledge, such as "*The President of the United States is Biden*" and "*The capital of the United States is Washington*", are often deeply entangled within model representations. Ensuring the invariance of such fine-grained irrelevant knowledge during parameter updates presents a significant challenge. To address this, we design a Knowledge Representation Disentanglement (KRD) module, which decomposes the subject representation of each target knowledge instance into two distinct components: one capturing target-knowledge-related information, and the other encoding target-knowledge-unrelated information.

Specifically, the KRD module consists of two components: the **Knowledge Disentangler** and the **Knowledge Recomposer**. For each edited knowledge triplet $(s, r, o)$, the Knowledge Disentangler takes the representations of the subject and relation as inputs and produces a pair of disentangled

vectors: the target-knowledge-related representation and the target-knowledge-unrelated representation. The Knowledge Recomposer then reconstructs the original representation from the disentangled components, ensuring consistency and completeness in the knowledge representation.

Formally, the target-knowledge-related representation $\mathbf{z}_e^r$ and target-knowledge-unrelated representation $\mathbf{z}_e^u$ for each edit sample $e = (s, r, o, o^*)$ are computed as:

$$\mathbf{z}_e^r = \mathrm{Dis}_r(\mathbf{h}_s, \mathbf{h}_r) = f(\mathbf{W}_1\mathbf{h}_s + \mathbf{W}_2\mathbf{h}_r), \quad \mathbf{z}_e^u = \mathrm{Dis}_u(\mathbf{h}_s, \mathbf{h}_r) = f(\mathbf{W}_3\mathbf{h}_s + \mathbf{W}_4\mathbf{h}_r), \quad (5)$$

where $f(\cdot)$ is the GELU Hendrycks & Gimpel (2016) activation function, and $\mathbf{W}_i \in \mathbb{R}^{d \times d}$ for $i = 1, \ldots, 4$ are trainable projection matrices.

The Knowledge Recomposer then reconstructs subject representation $\hat{\mathbf{h}}_s$ from $\mathbf{z}_e^u$ and $\mathbf{z}_e^r$:

$$\hat{\mathbf{h}}_s = \mathrm{Rec}(\mathbf{z}_e^r, \mathbf{z}_e^u) = \mathbf{W}_5\mathbf{z}_e^r + \mathbf{W}_6\mathbf{z}_e^u, \qquad (6)$$

where $\mathbf{W}_5$ and $\mathbf{W}_6 \in \mathbb{R}^{d \times d}$ are learnable parameters. The subject representation $\mathbf{h}_s$ and relation representation $\mathbf{h}_r$ are extracted from the hidden state of the last tokens in the subject $s$ and the prompt $p(s, r)$ at the $l$-th layer, respectively.

To ensure that the Knowledge Disentangler effectively separates the target-knowledge-related and the target-knowledge-unrelated representations, we introduce three complementary objectives:

**Knowledge Disentangling Loss** encourages the target-knowledge-related representation $\mathbf{z}_e^r$ and the target-knowledge-unrelated representation $\mathbf{z}_e^u$ to capture distinct attributes of the subject representation $\mathbf{h}_s$. To achieve this, we adopt a contrastive learning objective He et al. (2020) that maximize the mutual information (MI) between each of $\mathbf{z}_e^r$ and $\mathbf{h}_s$, and $\mathbf{z}_e^u$ and $\mathbf{h}_s$, while treating $\mathbf{z}_e^r$ and $\mathbf{z}_e^u$ as a negative pair to encourage their separation in the representation space.

Concretely, we define $(\mathbf{z}_e^r, \mathbf{h}_s)$ and $(\mathbf{z}_e^u, \mathbf{h}_s)$ as positive pairs, while treating $(\mathbf{z}_e^r, \mathbf{z}_e^u)$ as an additional negative pair. This lead to the following objective:

$$\mathcal{L}_{ctr} = \mathrm{InfoNCE}(\mathbf{z}_e^r, \mathbf{h}_s, [\mathbf{z}_e^u; \mathbf{H}_s]) + \mathrm{InfoNCE}(\mathbf{z}_e^u, \mathbf{h}_s, [\mathbf{z}_e^r; \mathbf{H}_s]), \qquad (7)$$

where $\mathbf{H}_s \in \mathbb{R}^{B \times d}$ represents $B$ negative subject representations sampled from the same batch. Here, $[\mathbf{z}_e^u; \mathbf{H}_s]$ and $[\mathbf{z}_e^r; \mathbf{H}_s]$ serve as the negative sample sets for $\mathbf{z}_e^r$ and $\mathbf{z}_e^u$, respectively. The InfoNCE loss is formulated as:

$$\mathrm{InfoNCE}(\mathbf{s}, \mathbf{s}^+, \mathbf{S}^-) = -\log \frac{\exp(\mathrm{sim}(\mathbf{s}, \mathbf{s}^+)/\tau)}{\sum_{\mathbf{s}' \in (\{\mathbf{s}^+\}, \mathbf{S}^-)} \exp(\mathrm{sim}(\mathbf{s}, \mathbf{s}')/\tau)}, \qquad (8)$$

where $\mathrm{sim}(\cdot, \cdot)$ refers to the cosine similarity function, and $\tau$ is the temperature parameter.

**Knowledge Constraint Loss** is designed to ensure that the disentangled representations effectively encode meaningful attribute of the subject $s$: the target-knowledge-related representation encodes the edited fact $(s, r, o)$, while the target-knowledge-unrelated representation preserves the fine-grained irrelevant facts $(s, r', o')$.

Specifically, for a given knowledge triple $(s, r, o)$, we sample a set of fine-grained irrelevant fact $\mathcal{N}$, where each $(s, r', o') \in \mathcal{N}$ shares the same subject $s$ but differs in relation and object. We construct corresponding prompts $p(s, r)$ and $p(s, r')$, which are separately fed into the LLM $F$. During forward computation, the original subject representation $\mathbf{h}_s$ is replaced with $\mathbf{z}_e^r$ and $\mathbf{z}_e^u$, respectively, to predict the target object $o$ and unrelated object $o'$. The loss is defined as:

$$\mathcal{L}_{con} = -\log P_{F(\mathbf{h}_s := \mathbf{z}_e^r)}[o|p(s, r)] + \sum_{(s, r', o') \in \mathcal{N}}^{|\mathcal{N}|} -\log P_{F(\mathbf{h}_s := \mathbf{z}_e^u)}[o'|p(s, r')], \qquad (9)$$

where $F(\mathbf{h}_s := \cdot)$ denotes the forward computation with the subject representation $\mathbf{h}_s$ replaced by the specified vector.

**Knowledge Reconstruction Loss** ensures that essential semantic information is retained during the disentanglement process by encouraging faithful reconstruction of the original subject representation. It is computed using the Mean Squared Error (MSE) between the original subject representation $\mathbf{h}_s$ and its reconstructed representation $\hat{\mathbf{h}}_s$:

$$\mathcal{L}_{recon} = \|\mathbf{h}_s - \hat{\mathbf{h}}_s\|_2. \qquad (10)$$

Finally, the parameters of the Knowledge Disentangler and Knowledge Recomposer are jointly optimized by minimizing the following overall objective:

$$\mathcal{L} = \mathcal{L}_{ctr} + \alpha \mathcal{L}_{con} + \beta \mathcal{L}_{recon}, \tag{11}$$

where $\alpha$ and $\beta$ are weighting coefficients that balance the contributions of different loss functions.

**Notably, the Knowledge Disentangler and Knowledge Recomposer, once trained, can be directly applied during the editing process without requiring retraining for each individual edit sample.** Detailed training procedures are provided in Appendix F.1.

## 3.2 DISENTANGLEMENT-BASED KNOWLEDGE EDIT

In this section, we discuss how to leverage disentangled representations to perform parameter updates for injecting target knowledge, while explicitly preserving fine-grained irrelevant knowledge.

Following MEMIT Meng et al. (2023), we formulate the editing process as the injection of key-value pairs into the FFN. For each edit instance $(s, r, o^*)$, we compute a key-value pair $(\mathbf{k}_*, \mathbf{v}_*)$ that guide the parameter update. The key $\mathbf{k}_*$ is computed by averaging across $N$ randomly generated prefixes attached prompts:

$$\mathbf{k}_* = \frac{1}{N} \sum\nolimits_{j=1}^{N} f(\mathbf{W}_{in}^l \cdot \mathbf{h}_s^{l-1}). \tag{12}$$

**Target Knowledge Editing**. To compute the corresponding value $\mathbf{v}_*$, we first disentangle the subject representation $\mathbf{h}_s$ into $\mathbf{z}_e^r$ and $\mathbf{z}_e^u$. Unlike MEMIT and ROME, which directly optimize the FFN output at a specific layer to encode the new knowledge $(s, r, o^*)$, our approach instead optimizes a modification $\delta$ to the target-knowledge-related representation $\mathbf{z}_e^r$:

$$\begin{aligned} \mathbf{h}_s^* &= \text{Rec}(\mathbf{z}_e^r + \delta, \mathbf{z}_e^u), \\ \delta &= \arg\min_{\delta} \; -\log \mathrm{P}_{F(\mathbf{h}_s := \mathbf{h}_s^*)} \left[ o^* \mid p(s, r) \right]. \end{aligned} \tag{13}$$

Based on the residual formulation of hidden states in Equation (1), the updated subject representation $\mathbf{h}_s^*$ at the editing layer can be expressed as:

$$\mathbf{h}_s^* = \mathbf{h}_s^0 + \sum\nolimits_{l=1}^{L} \mathbf{a}_s^l + \sum\nolimits_{l=1}^{L-1} \mathbf{v}_s^l + \mathbf{v}_*, \tag{14}$$

where $\mathbf{h}_s^0$ is the initial embedding of the subject, and $\mathbf{a}_s^l$ and $\mathbf{v}_s^l$ denotes the output of the attention module and FFN at layer $l$, respectively.

Consequently, we compute the value $\mathbf{v}_*$ to be injected at the FFN layer as:

$$\mathbf{v}_* = \mathbf{h}_s^* - \mathbf{h}_s^p, \quad \text{where} \quad \mathbf{h}_s^p = \mathbf{h}_s^0 + \sum\nolimits_{l=1}^{L} \mathbf{a}_s^l + \sum\nolimits_{l=1}^{L-1} \mathbf{v}_s^l. \tag{15}$$

**Fine-grained Irrelevant Knowledge Preserving**. To prevent unintended modifications to fine-grained irrelevant knowledge during the editing process, we enforce that the disentangled unrelated representation remains invariant before and after editing. This is formalized as minimizing:

$$\begin{aligned} \|\text{Dis}_u(\mathbf{h}_s^*, \mathbf{h}_r) - \text{Dis}_u(\mathbf{h}_s, \mathbf{h}_r)\|_F^2 &= \left\| \text{Dis}_u(\mathbf{h}_s^p + \hat{\mathbf{W}}\mathbf{k}_*, \mathbf{h}_r) - \text{Dis}_u(\mathbf{h}_s, \mathbf{h}_r) \right\|_F^2 \\ &= \left\| f\left( \mathbf{W}_3(\mathbf{h}_s^p + \hat{\mathbf{W}}\mathbf{k}_*) + \mathbf{W}_4\mathbf{h}_r \right) - f\left( \mathbf{W}_3\mathbf{h}_s + \mathbf{W}_4\mathbf{h}_r \right) \right\|_F^2. \end{aligned}$$

Here, $\hat{\mathbf{W}}$ represents the updated weights, $\mathbf{h}_s^p + \hat{\mathbf{W}}\mathbf{k}^*$ denotes the subject representation extracted by the edited model. For simplicity, we omit the activation function and enforce consistency in the representation before activation, leading to the following constraint:

$$\left\| \mathbf{W}_3 \left( \mathbf{h}_s^p + \hat{\mathbf{W}}\mathbf{k}_* \right) - \mathbf{W}_3\mathbf{h}_s \right\|_F^2 = \left\| \mathbf{W}_3 \left( \hat{\mathbf{W}}\mathbf{k}_* - \mathbf{v}_0 \right) \right\|_F^2, \tag{16}$$

where $\mathbf{v}_0 = \mathbf{h}_s - \mathbf{h}_s^p = \mathbf{W}\mathbf{k}_*$ is the original output of the edited FFN module.

While MEMIT and ROME preserve existing knowledge by maintaining the correspondence between $\mathbf{K}_0$ and $\mathbf{V}_0$, as described in Section 2.2, our approach further ensures that the disentangled target-knowledge-unrelated representations of knowledge set $(\mathbf{K}_0, \mathbf{V}_0)$ remain unchanged before and after

editing. To achieve this, we minimize the following objective function:

$$\sum_{i=1}^{|\mathbf{K}_0|} \left\| \text{Dis}_u(\mathbf{h}_{s_i}^p + \hat{\mathbf{W}}\mathbf{k}_i, \mathbf{h}_{r_i}) - \text{Dis}_u(\mathbf{h}_{s_i}, \mathbf{h}_{r_i}) \right\|_F^2 \Rightarrow \left\| \mathbf{W}_3(\hat{\mathbf{W}}\mathbf{K}_0 - \mathbf{V}_0) \right\|_F^2, \quad (17)$$

where $\mathbf{k}_i \in \mathbf{K}_0$, $\mathbf{h}_{s_i}$ and $\mathbf{h}_{r_i}$ denote the corresponding representations of subject $s_i$ and $r_i$ in the edited layer. The full derivation is provided in Appendix C.1.

**Rank-One Parameter Update**. By composing Equations (3),(16), and (17), we derive the final parameter updating objective:

$$\hat{\mathbf{W}} = \underset{\hat{\mathbf{W}}}{\arg\min} \Bigg( \underbrace{\left\| \hat{\mathbf{W}}\mathbf{k}_* - \mathbf{v}_* \right\|_F^2}_{\text{Target knowledge editing}} + \underbrace{\left\| \hat{\mathbf{W}}\mathbf{K}_0 - \mathbf{V}_0 \right\|_F^2}_{\text{Coarse-grained irrelevant knowledge preserving}}$$

$$+ \underbrace{\left\| \mathbf{W}_3 \left( \hat{\mathbf{W}}\mathbf{k}_* - \mathbf{v}_0 \right) \right\|_F^2 + \left\| \mathbf{W}_3 \left( \hat{\mathbf{W}}\mathbf{K}_0 - \mathbf{V}_0 \right) \right\|_F^2}_{\text{Fine-grained irrelevant knowledge preserving}} \Bigg). \quad (18)$$

The closed-form solution to this optimization problem is:

$$\hat{\mathbf{W}} = \mathbf{W} + (\mathbf{W}_3^T\mathbf{W}_3 + \mathbf{E})^{-1}\mathbf{\Delta}_{\text{MEMIT}}, \quad (19)$$

where $\mathbf{E}$ is the identity matrix and $\mathbf{\Delta}_{\text{MEMIT}}$ represents the parameter update derived from MEMIT (Equation (4)). The detailed derivation is provided in Appendix C.2.

## 4 EXPERIMENTS

In this section, we first introduce our newly constructed dataset, FINE-KED, designed to evaluate the impact of editing on fine-grained irrelevant knowledge. To provide a more comprehensive assessment of our DiKE's performance in gereral editing scenarios, we further evaluate DiKE on the widely used COUNTERFACT and MQUAKE-3K benchmarks across GPT2-XL (1.5B), GPT-J (6B), and LLaMA3 (8B). More detailed experimental results are provided in Appendix G.

### 4.1 FINE-KED DATASET

FINE-KED is designed to evaluate the impact of editing methods on fine-grained irrelevant knowledge. For each edit sample $(s, r, o)$, we construct fine-grained irrelevant knowledge $(s, r', o')$ and categorize the dataset into three levels: **Easy**, **Middle**, and **Hard**, based on the semantic relatedness between relation $r$ and $r'$ (examples of these levels are provided in Table 1). We employ *Efficacy* to quantify the success rate of edits and *Relational Locality* to assess the preservation of fine-grained irrelevant knowledge. Detailed information is provided in Appendix D.1.

Table 1: Relation Examples in Different Levels of FINE-KED

| Level | Relations |
|---|---|
| Easy | The name of the child of {} is
The name of the award {} won is |
| Middle | The place of death of {} is
The place of birth of {} is |
| Hard | The name of the head of state of {} is
The name of the capital city of {} is |

### 4.2 EXPERIMENTAL SETUPS

**Datasets and Evaluation Metrics.** In addition to FINE-KED, we conduct experiments on COUNTERFACT (Meng et al., 2022), a benchmark designed to evaluate the insertion of counterfactual knowledge into LLMs. Performance is assessed using three key metrics: *Efficacy Score*, *Paraphrase Score*, and *Neighborhood Score*. Further details are in Appendix D.2. To further investigate DiKE's generalization capability in more complex scenarios, we also evaluate on the multi-hop reasoning dataset MQUAKE-3K Zhong et al. (2023). Dataset descriptions are provided in Appendix D.3, with corresponding results reported in Appendix G.2.

Table 2: Performance comparison on FINE-KED in terms of Efficacy (%) and Relational Locality (%). The best performance is highlighted in **boldface**, and the second-best is underlined.

| Method | Eff. | R-Loc. (GPT2-XL) | | | | Eff. | R-Loc. (GPT-J) | | | | Eff. | R-Loc. (LLaMA-3) | | | |
|---|---|---|---|---|---|---|---|---|---|---|---|---|---|---|---|
| | | Easy | Mid. | Hard | Avg. | | Easy | Mid. | Hard | Avg. | | Easy | Mid. | Hard | Avg. |
| BASE | 20.4 | 62.7 | 65.7 | 71.9 | 65.8 | 21.7 | 86.9 | 87.6 | 89.2 | 87.6 | 23.9 | 81.1 | 80.3 | 81.3 | 80.9 |
| FT | **99.1** | 46.8 | 45.0 | 40.9 | 44.9 | **100** | 69.2 | 64.8 | 60.6 | 65.9 | **100** | 69.4 | 63.0 | 49.6 | 62.8 |
| MEND | 94.2 | 47.1 | 47.5 | 31.5 | 43.4 | 98.7 | 71.7 | 66.5 | 51.8 | 65.4 | 94.3 | 71.5 | 63.4 | 54.7 | 65.2 |
| ROME | 91.8 | 48.3 | 45.4 | 56.3 | 49.5 | 99.9 | 53.5 | 49.4 | 44.9 | 50.3 | 99.8 | 56.8 | 50.7 | 47.8 | 53.0 |
| ROME-C | 91.7 | 49.7 | 46.7 | 57.4 | 50.8 | 99.9 | 63.3 | 58.3 | 59.4 | 61.0 | 99.9 | 65.5 | 57.9 | 60.9 | 62.3 |
| MEMIT | 90.8 | 49.0 | 45.0 | 55.9 | 49.6 | 99.9 | 61.9 | 56.0 | 59.5 | 59.7 | 98.7 | 64.6 | 54.9 | 62.6 | 61.5 |
| MEMIT-C | 91.0 | 49.5 | 45.5 | 55.3 | 49.8 | 99.8 | 67.3 | 62.3 | 67.1 | 65.9 | 97.2 | 68.1 | 60.3 | 69.2 | 66.3 |
| AlphaEdit | 98.7 | 47.3 | 43.9 | 47.4 | 46.4 | 99.9 | 59.8 | 54.9 | 54.8 | 57.2 | 98.2 | 68.1 | 59.2 | 67.8 | 65.6 |
| **DiKE** | 97.4 | **54.2** | **51.8** | **60.2** | **55.0** | 99.1 | **72.0** | **67.4** | **74.1** | **71.3** | 99.1 | **72.7** | **65.3** | **72.4** | **70.6** |
| **Improve** | - | 9.1% | 9.1% | 4.9% | 8.3% | - | 0.4% | 1.4% | 10.4% | 8.2% | - | 1.7% | 3.0% | 4.6% | 6.5% |

**Baselines.** Our experiments are conducted on three LLMs: GPT2-XL (1.5B) (Radford et al., 2019), GPT-J (6B) (Wang & Komatsuzaki, 2021) and LLaMA3 (8B) (Dubey et al., 2024). We compare our method with a number of knowledge editing methods: Fine-Tuning (FT) (Zhu et al., 2020), MEND (Mitchell et al., 2022), ROME (Meng et al., 2022), MEMIT (Meng et al., 2023), and AlphaEdit (Fang et al., 2025). To further validate the superiority of DiKE in preserving the fine-grained irrelevant knowledge, we compare it with two variant models ROME-C and MEMIT-C on FINE-KED, which directly incorporate multiple additional relational constraints $(s, r_i, o_i)$ into the LLM in each editing. The implementation details of baselines and DiKE in Appendix F.1.

## 4.3 EXPERIMENT RESULTS

Through these experiments, we aim to address the following key research questions:

**How does DiKE Perform in Preserving Fine-grained knowledge?** To avoid performance inflation due to data leakage and to verify the generalization capability of our KRD module, we ensure minimal subject overlap between the KRD training data and the evaluation datasets. Specifically, the subject overlap rates between the KRD module's training set and the edited samples in FINE-KED and COUNTERFACT are only 1.39% and 6.33%, respectively. A detailed investigation of the generalization behavior of the KRD module is provided in Appendix G.6.

**(i) Performance on FINE-KED.** Table 2 presents the performance of all editors on FINE-KED. From the results, we can draw the following observations: (1) DiKE excels in all evaluations of Relational Locality while maintaining high editing success rate. Specifically, DiKE performs enhancements up to 8.3% on the Relational Locality metric over the baseline model. This demonstrates that fine-grained disentanglement and constraint of target knowledge can effectively mitigate negative impacts on fine-grained unrelated knowledge without compromising editing performance. (2) ROME, MEMIT and AlphaEdit generally perform worse on the Relational Locality compared to DiKE. MEND and FT, demonstrate better performance than ROME and MEMIT on the Easy and Middle levels of FINE-KED. However, their performance declines significantly on the Hard level of the dataset. We believe this is because these models directly manipulate the subject or relation representation to perform parameter updates, which can easily have negative impacts on fine-grained unrelated knowledge that is entangled within the parameters and representations. Although AlphaEdit introduces null-space constraints to preserve unrelated knowledge, the constraint is applied at a coarse-grained level and thus struggles to prevent interference with fine-grained irrelevant knowledge. (3) ROME-C and MEMIT-C, which incorporate additional relational constraints, improve upon their original versions but still fall short of DiKE. This suggests that merely adding constraint samples is insufficient to reliably preserve fine-grained unrelated knowledge. Furthermore, the requirement to manually construct extra constraints for each editing instance introduces substantial overhead. In contrast, DiKE, with its KRD moudle, requires only one-time training process and can be directly applied during subsequent edits without retraining, significantly enhancing editing efficiency.

**(ii) Performance on COUNTERFACT.** Table 3 presents the performance of all editors on COUNTERFACT. The results show that DiKE achieves competitive results across other key editing evaluation metrics, such as Paraphrase Score and Neighborhood Score.

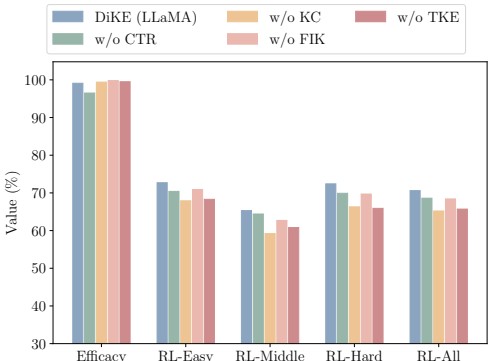

Figure 3: Ablation studies on FINE-KED in terms of Efficacy and Relational Locality.

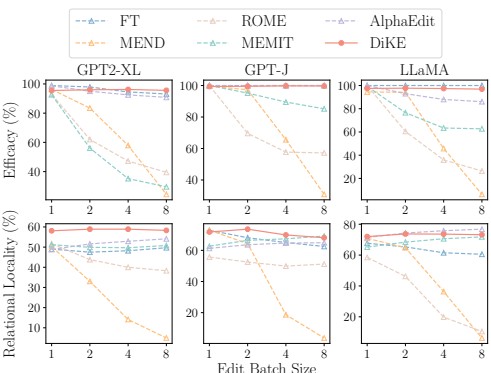

Figure 4: Performance of subject-consistent batch editing on the FINE-KED.

These findings suggest that the disentanglement-based approach, DiKE, not only effectively preserves fine-grained irrelevant knowledge but also maintains strong generalization ability and better retention of unrelated neighborhood knowledge.

**How do Different Components Affect the DiKE Performance?** To investigate the effectiveness of each component in DiKE, we conduct an ablation study on LLaMA3 using the following variants: w/o CTR, which removes knowledge disentangling loss from the KRD module; w/o KC, which excludes knowledge constraint loss from KRD module; w/o TKE, which performs editing directly on the original representations rather than the disentangled ones; and w/o FIK, which removes the constraint for preserving fine-grained irrelevant knowledge in the DKE module. Figure 3 presents the experimental results on FINE-KED, leading to the following conclusions: (1) DiKE outperforms w/o CTR and w/o KC on most Relational Locality metrics, confirming that contrastive learning and knowledge constraint

Table 3: Performance comparison on COUNTERFACT in terms of Efficacy Score, Paraphrase Score, and Neighborhood Score. The Avg. is the harmonic mean of the three metrics.

| Model | Method | Avg. | Effi. | Para. | Neigh. |
|---|---|---|---|---|---|
| GPT-J (6B) | BASE | 23.8 | 16.2 | 19.2 | 81.0 |
| | FT | 24.4 | **100.0** | 95.8 | 9.8 |
| | MEND | 62.2 | 97.7 | 53.2 | 52.1 |
| | ROME | 90.3 | 99.9 | **99.7** | 75.9 |
| | MEMIT | 90.4 | 99.6 | 95.2 | 79.2 |
| | AlphaEdit | **91.0** | 99.6 | 96.9 | **79.3** |
| | **DiKE** | 90.8 | 99.8 | 96.1 | **79.3** |
| LLaMA3 (8B) | BASE | 14.1 | 9.2 | 10.9 | 85.9 |
| | FT | 18.4 | **100.0** | 97.7 | 7.0 |
| | MEND | 58.4 | **100.0** | 73.6 | 36.0 |
| | ROME | 90.8 | **100.0** | **98.9** | 77.3 |
| | MEMIT | 91.1 | 99.8 | 94.6 | 80.9 |
| | AlphaEdit | 89.7 | **100.0** | 93.5 | 78.5 |
| | **DiKE** | **92.4** | 99.9 | 96.6 | **82.8** |

contribute to effective disentanglement. (2) Compared with DiKE, w/o TKE exhibits a substantial performance drop, underscoring the importance of disentangling and editing the target-knowledge-related representation rather than the original entangled one. (3) Compared with w/o FIK, DiKE further improves on Relational Locality, especially in middle and hard level, demonstrating the benefit of explicitly enforcing invariance on the disentangled target-knowledge-unrelated representation. Experimental results on GPT2-XL and GPT-J are shown in Figure 8.

**Which Scenarios Benefit from the DiKE Approach?** To further validate the importance of representation-disentanglement-based editing in preserving fine-grained irrelevant knowledge, we conduct a **subject-consistent batch editing** experiment where all edit samples within a batch share the same subject. This setup exerts a stronger influence on the subject's representation, making the task significantly more challenging than conventional batch editing settings, in which edit samples are typically unrelated. We evaluate performance on FINE-KED under batch sizes of 1, 2, 4, and 8, with results shown in Figure 4.

We observe that DiKE achieves an Efficacy Score close to 100% across most models and batch settings, while maintaining the highest Relational Locality Score. By contrast, ROME exhibits a sharp degradation as batch size increases. Although MEMIT and AlphaEdit incorporate mechanisms for batch editing and irrelevant knowledge preservation, respectively, their improvements in Relational Locality under larger batch settings remain significantly lower than those of DiKE. This advantage arises because DiKE explicitly disentangles subject representations across different relations, thereby isolating relation-specific knowledge and minimizing interference. As a result,

edits to one relation have little impact on other knowledge associated with the same subject. By contrast, ROME, MEMIT, and AlphaEdit directly update entangled subject representations, where overlapping relational information leads to conflicts, imprecise updates, and reduced effectiveness. Furthermore, AlphaEdit constrains only coarse-grained irrelevant knowledge, making it less effective at preserving fine-grained relational distinctions within subject representations. We also find that MEND suffers from a significant performance drop in this setting, which may be attributed to its reliance on a hypernetwork to predict parameter updates based on gradients. When editing a batch of knowledge instances sharing the same subject, the predicted gradients are more likely to conflict with each other, resulting in editing failures.

## 5  RELATED WORK

In this section, we introduce the related work on knowledge editing, which aims to inject new knowledge into LLMs or modify their internal knowledge, while minimizing unintended changes to unrelated knowledge. This study focuses on parameter-modifying methods, which can be broadly categorized into three groups:

**Fine-tuning-based methods** (Gu et al., 2024; Yu et al., 2024; Ni et al., 2024) utilize efficient parameter-tuning techniques to update model knowledge. To alleviate issues such as overfitting, these methods typically introduce additional constraints to preserve unrelated knowledge. For example, RECT (Gu et al., 2024) injects new knowledge by selecting and fine-tuning the top-$k$ parameters most relevant to the target, while simultaneously constraining the magnitude of updates to reduce interference with other knowledge.

**Meta-learning-based methods** employ a hypernetwork to generate editing-specific parameter updates. MEND (Mitchell et al., 2022) uses a low-rank gradient decomposition and a lightweight hypernetwork to transform fine-tuning gradients into weight updates. MALMEN (Tan et al., 2024) extends MEND to batch editing by formulating the update process as a least-squares optimization problem.

**Locate-then-edit Methods** perform editing by first identifying model parameters associated with the target knowledge and then applying targeted updates (Meng et al., 2023; Gupta et al., 2024; Zhang et al., 2024; Fang et al., 2025; Zhang et al., 2025b;a). Early work such as Knowledge Neurons (Dai et al., 2022), proposed a knowledge attribution method to identify relevant neurons. However, this approach exhibits limitations in precisely adjusting model weights. Subsequently, ROME (Meng et al., 2022) treats the weights of the FFN layers as a form of linear associative memory and updates specific layers to encode new knowledge. MEMIT (Meng et al., 2023) extends ROME by enabling large-scale knowledge editing through shared updates across multiple layers, thereby mitigating interference with previously edited layers. AlphaEdit (Fang et al., 2025) extends this line of work by introducing null-space constraints to prevent updates from affecting unrelated knowledge. However, its constraints are applied at a coarse-grained level, making it less effective at preserving fine-grained irrelevant knowledge that may be entangled with the target in the representation space.

## 6  CONCLUSION

In this paper, we have proposed DiKE for knowledge editing in LLMs. DiKE leverages a Knowledge Representation Disentanglement module to separate target-related knowledge from fine-grained unrelated knowledge. Building on this, we introduce a Disentanglement-based Knowledge Edit module that injects the edited knowledge while preserving the fine-grained neighboring irrelevant knowledge. Experimental results across three LLMs on our constructed FINE-KED and COUNTERFACT, demonstrate the effectiveness and superiority of DiKE in model editing tasks.

### ACKNOWLEDGEMENTS

This work was supported by the National Natural Science Foundation of China (Nos. 62502286, 62472261, 62576339, 62372275). It was also Funded by Basic Research Program of Jiangsu (BK20250429), the Shandong Provincial Natural Science Foundation (ZR2024QF203), the Technology Innovation Guidance Program of Shandong Province (YDZX2024088), the Provincial Key

R&D Program of Shandong Province (2024CXGC010108), and the Open Research Fund of the State Key Laboratory of Multimodal Artificial Intelligence Systems.

## ETHICS STATEMENT

The goal of this work is to advance the field of knowledge editing in LLMs, aiming to improve the precision and reliability of information they contain. Our proposed method, DiKE, is designed to update factual knowledge while minimizing unintended side effects on related information, which we believe is a step toward more controllable and trustworthy AI systems. Such improvements have positive applications in areas like correcting factual inaccuracies, removing harmful or biased content, and keeping models updated with the latest information without costly retraining.

However, we acknowledge that like any technology that allows for the modification of information, knowledge editing tools could be misused. A malicious actor could potentially leverage such techniques to insert subtle misinformation, propaganda, or harmful biases into an LLM. While our work focuses on improving the technical fidelity of edits, it does not in itself prevent such applications. We believe the broader research community must continue to develop robust detection mechanisms and safeguards in parallel with advancements in editing techniques. We encourage the responsible development and deployment of knowledge editing technologies, with a strong emphasis on ethical considerations and safeguards against misuse.

## REPRODUCIBILITY STATEMENT

To ensure the reproducibility of our results, we provide detailed descriptions of our methodology and experimental setup throughout the paper. All experiments were conducted using publicly available Large Language Models: GPT2-XL (1.5B), GPT-J (6B), and LLaMA3 (8B). We evaluate our method on the widely-used COUNTERFACT and MQUAKE-3K benchmarks. For evaluating the preservation of fine-grained irrelevant knowledge, we introduce a new benchmark named FINE-KED, and we will release the dataset publicly upon publication.

Appendix F.1 provides specific implementation details, including key hyperparameters such as learning rates, batch sizes, optimizer settings, and the specific layers targeted for editing in each model. For baseline comparisons, we utilized the experimental framework provided by Meng et al. (2023) to ensure consistency. To further aid reproducibility, we will make our source code and the newly constructed FINE-KED dataset publicly available.

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

## A    THE USE OF LARGE LANGUAGE MODELS

During the preparation of this work, we use a LLM for assistance with the following tasks: (1) refining the language and improving the clarity of the manuscript, particularly in the Methodology and Results sections; (2) correcting grammatical errors and ensuring stylistic consistency.

## B    LIMITATIONS & FUTURE DISCUSSION

We acknowledge several limitations of our work, which suggest promising directions for future research.

The first limitation of DiKE is its focus on structured knowledge editing tasks, similar to those addressed by ROME, MEMIT, and AlphaEdit. Specifically, DiKE is best suited for scenarios where knowledge can be explicitly represented as relational triples $(s, r, o)$. This design choice may limit its applicability in broader knowledge editing contexts, especially those involving unstructured or semi-structured knowledge formats. Extending disentangled representation learning and editing to support more diverse and unstructured forms of knowledge remains an important direction for future work.

Second, the scope of our evaluation benchmark, FINE-KED, is currently limited to measuring editing success and the preservation of fine-grained irrelevant knowledge. While it provides a targeted assessment of editing success and preservation of fine-grained irrelevant knowledge, these dimensions alone are insufficient to fully characterize the broader capabilities required in knowledge editing. In particular, many real-world editing scenarios involve additional challenges such as generalization to paraphrases, multi-hop reasoning, and robustness to distribution shifts. Therefore, FINE-KED should be viewed as a complementary benchmark rather than a standalone evaluation. In future work, we plan to further expand FINE-KED to better capture the diverse challenges in model editing and provide a more complete assessment of editing performance.

Finally, we focus specifically on fine-grained irrelevant knowledge that shares the same subject as the edited fact in this study. This choice is motivated by the observation that existing representative editing methods, such as AlphaEdit and ROME, perform poorly on this type of knowledge (as shown in Fig.1), highlighting it as a particularly challenging and underexplored problem. We believe that addressing this category is both meaningful and necessary. Moreover, in structured knowledge editing scenarios, defining fine-grained irrelevant knowledge via shared subjects but differing relations is both intuitive and formally tractable, which facilitates evaluation and methodological design. In future work, we plan to explore other types of fine-grained irrelevant knowledge beyond subject overlap to further broaden the scope and generality of our framework.

## C    DERIVATION DETAILS FOR FINE-GRAINED IRRELEVANT KNOWLEDGE PRESERVATION

### C.1    DERIVATION OF EQUATION (17)

We provide a detailed derivation of Equation equation 17 from Section 3.2, which enforces the consistency of the disentangled target-knowledge-unrelated representations before and after editing:

$$\sum_{i=1}^{|\mathbf{K}_0|} \left\| \text{Dis}_u(\mathbf{h}_{s_i}^p + \hat{\mathbf{W}}\mathbf{k}_i, \mathbf{h}_{r_i}) - \text{Dis}_u(\mathbf{h}_{s_i}, \mathbf{h}_{r_i}) \right\|_F^2 \ \Rightarrow \ \left\| \mathbf{W}_3(\hat{\mathbf{W}}\mathbf{K}_0 - \mathbf{V}_0) \right\|_F^2 .$$

We begin by expanding the definition of Knowledge Disentangler:

$$\begin{aligned}
&\sum_{i=1}^{|\mathbf{K}_0|} \left\| \text{Dis}_u\left( \mathbf{h}_{s_i}^p + \hat{\mathbf{W}}\mathbf{k}_i, \mathbf{h}_{r_i} \right) - \text{Dis}_u\left( \mathbf{h}_{s_i}, \mathbf{h}_{r_i} \right) \right\|_F^2 \\
&= \sum_{i=1}^{|\mathbf{K}_0|} \left\| f\left( \mathbf{W}_3\left( \mathbf{h}_{s_i}^p + \hat{\mathbf{W}}\mathbf{k}_i \right) + \mathbf{W}_4\mathbf{h}_{r_i} \right) - f\left( \mathbf{W}_3\mathbf{h}_{s_i} + \mathbf{W}_4\mathbf{h}_{r_i} \right) \right\|_F^2 .
\end{aligned} \tag{20}$$

Following the approximation in Equation (16), we omit the nonlinearity activation function and enforce consistency at the pre-activation level:

$$
\begin{aligned}
&\sum_{i=1}^{|\mathbf{K}_0|} \left\| \left( \mathbf{W}_3(\mathbf{h}_{s_i}^p + \hat{\mathbf{W}}\mathbf{k}_i) + \mathbf{W}_4\mathbf{h}_{r_i} \right) - (\mathbf{W}_3\mathbf{h}_{s_i} + \mathbf{W}_4\mathbf{h}_{r_i}) \right\|_F^2 \\
&= \sum_{i=1}^{|\mathbf{K}_0|} \left\| \mathbf{W}_3(\mathbf{h}_{s_i}^p + \hat{\mathbf{W}}\mathbf{k}_i - \mathbf{h}_{s_i}) \right\|_F^2 \\
&= \sum_{i=1}^{|\mathbf{K}_0|} \left\| \mathbf{W}_3(\hat{\mathbf{W}}\mathbf{k}_i - \mathbf{v}_i) \right\|_F^2 \\
&= \sum_{i=1}^{|\mathbf{K}_0|} \mathrm{Tr}\left( \mathbf{W}_3(\hat{\mathbf{W}}\mathbf{k}_i - \mathbf{v}_i)(\hat{\mathbf{W}}\mathbf{k}_i - \mathbf{v}_i)^\top \mathbf{W}_3^\top \right) \\
&= \mathrm{Tr}\left( \sum_{i=1}^{|\mathbf{K}_0|} \mathbf{W}_3(\hat{\mathbf{W}}\mathbf{k}_i - \mathbf{v}_i)(\hat{\mathbf{W}}\mathbf{k}_i - \mathbf{v}_i)^\top \mathbf{W}_3^\top \right) \\
&= \mathrm{Tr}\left( \mathbf{W}_3(\hat{\mathbf{W}}\mathbf{K}_0 - \mathbf{V}_0)(\hat{\mathbf{W}}\mathbf{K}_0 - \mathbf{V}_0)^\top \mathbf{W}_3^\top \right) \\
&= \left\| \mathbf{W}_3(\hat{\mathbf{W}}\mathbf{K}_0 - \mathbf{V}_0) \right\|_F^2,
\end{aligned}
\tag{21}
$$

where $\mathrm{Tr}(\cdot)$ denotes the trace operator.

## C.2 DERIVATION OF THE CLOSED-FORM SOLUTION FOR EQUATION (18)

We provide a detailed derivation of the closed-form solution for the optimization problem defined in Equation (18) from Section 3.2:

$$
\hat{\mathbf{W}} = \underset{\hat{\mathbf{W}}}{\mathrm{argmin}} \Bigg( \underbrace{\left\| \hat{\mathbf{W}}\mathbf{k}_* - \mathbf{v}_* \right\|_F^2}_{\text{Target knowledge editing}} + \underbrace{\left\| \hat{\mathbf{W}}\mathbf{K}_0 - \mathbf{V}_0 \right\|_F^2}_{\text{Coarse-grained irrelevant knowledge preserving}}
$$
$$
+ \underbrace{\left\| \mathbf{W}_3\left( \hat{\mathbf{W}}\mathbf{k}_* - \mathbf{v}_0 \right) \right\|_F^2 + \left\| \mathbf{W}_3\left( \hat{\mathbf{W}}\mathbf{K}_0 - \mathbf{V}_0 \right) \right\|_F^2}_{\text{Fine-grained irrelevant knowledge preserving}} \Bigg).
$$

To start, we denote the update as $\hat{\mathbf{W}}$ with $\mathbf{W} + \Delta\mathbf{W}$, and reformulate the objective as:

$$
\begin{aligned}
L(\Delta\mathbf{W}) =\ & \|(\mathbf{W} + \Delta\mathbf{W})\mathbf{k}_* - \mathbf{v}_*\|_F^2 + \|\mathbf{W}_3\left((\mathbf{W} + \Delta\mathbf{W})\mathbf{k}_* - \mathbf{v}_0\right)\|_F^2 \\
& + \|(\mathbf{W} + \Delta\mathbf{W})\mathbf{K}_0 - \mathbf{V}_0\|_F^2 + \|\mathbf{W}_3\left((\mathbf{W} + \Delta\mathbf{W})\mathbf{K}_0 - \mathbf{V}_0\right)\|_F^2 \\
=\ & \|\Delta\mathbf{W}\mathbf{k}_* - (\mathbf{v}_* - \mathbf{W}\mathbf{k}_*)\|_F^2 + \|\mathbf{W}_3\Delta\mathbf{W}\mathbf{k}_* - \mathbf{W}_3(\mathbf{v}_0 - \mathbf{W}\mathbf{k}_*)\|_F^2 \\
& + \|\Delta\mathbf{W}\mathbf{K}_0 - (\mathbf{V}_0 - \mathbf{W}\mathbf{K}_0)\|_F^2 + \|\mathbf{W}_3\Delta\mathbf{W}\mathbf{K}_0 - \mathbf{W}_3(\mathbf{V}_0 - \mathbf{W}\mathbf{K}_0)\|_F^2.
\end{aligned}
\tag{22}
$$

To facilitate the derivation, we recall a general form of Frobenius norm minimization:

$$
\begin{aligned}
\hat{L}(\mathbf{W}) &= \|\mathbf{A}\mathbf{W}\mathbf{B} - \mathbf{C}\|_F^2 \\
&= \mathrm{Tr}\left( (\mathbf{A}\mathbf{W}\mathbf{B} - \mathbf{C})(\mathbf{A}\mathbf{W}\mathbf{B} - \mathbf{C})^\top \right) \\
&= \mathrm{Tr}\left( \mathbf{A}\mathbf{W}\mathbf{B}\mathbf{B}^\top\mathbf{W}^\top\mathbf{A}^\top - \mathbf{A}\mathbf{W}\mathbf{B}\mathbf{C}^\top - \mathbf{C}\mathbf{B}^\top\mathbf{W}^\top\mathbf{A}^\top + \mathbf{C}\mathbf{C}^\top \right).
\end{aligned}
\tag{23}
$$

Next, we compute the gradient of $\hat{L}(\mathbf{W})$ with respect to $\mathbf{W}$:

$$
\begin{aligned}
\nabla_{\mathbf{W}}\hat{L}(\mathbf{W}) &= \nabla_{\mathbf{W}}\operatorname{Tr}\left(\mathbf{AWBB}^\top\mathbf{W}^\top\mathbf{A}^\top - \mathbf{AWBC}^\top - \mathbf{CB}^\top\mathbf{W}^\top\mathbf{A}^\top + \mathbf{CC}^\top\right) \\
&= \frac{\partial}{\partial\mathbf{W}}\operatorname{Tr}\left(\mathbf{AWBB}^\top\mathbf{W}^\top\mathbf{A}^\top\right) - \frac{\partial}{\partial\mathbf{W}}\operatorname{Tr}\left(\mathbf{AWBC}^\top\right) \\
&\quad - \frac{\partial}{\partial\mathbf{W}}\operatorname{Tr}\left(\mathbf{CB}^\top\mathbf{W}^\top\mathbf{A}^\top\right) + \frac{\partial}{\partial\mathbf{W}}\operatorname{Tr}\left(\mathbf{CC}^\top\right) \\
&= 2\left(\mathbf{A}^\top\mathbf{AWBB}^\top - \mathbf{A}^\top\mathbf{CB}^\top\right).
\end{aligned}
\tag{24}
$$

This result serves as the foundation for deriving the gradient for $L(\Delta\mathbf{W})$ with respect to $\Delta\mathbf{W}$:

$$
\begin{aligned}
\nabla_{\Delta\mathbf{W}}L(\Delta\mathbf{W}) &= 2\left(\Delta\mathbf{W}\mathbf{k}_*\mathbf{k}_*^\top - (\mathbf{v}_* - \mathbf{W}\mathbf{k}_*)\mathbf{k}_*^\top\right) \\
&\quad + 2\left(\mathbf{W}_3^\top\mathbf{W}_3\Delta\mathbf{W}\mathbf{k}_*\mathbf{k}_*^\top - \mathbf{W}_3^\top\mathbf{W}_3(\mathbf{v}_0 - \mathbf{W}\mathbf{k}_*)\mathbf{k}_*^\top\right) \\
&\quad + 2\left(\Delta\mathbf{W}\mathbf{K}_0\mathbf{K}_0^\top - (\mathbf{V}_0 - \mathbf{W}\mathbf{K}_0)\mathbf{K}_0^\top\right) \\
&\quad + 2\left(\mathbf{W}_3^\top\mathbf{W}_3\Delta\mathbf{W}\mathbf{K}_0\mathbf{K}_0^\top - \mathbf{W}_3^\top\mathbf{W}_3(\mathbf{V}_0 - \mathbf{W}\mathbf{K}_0)\mathbf{K}_0^\top\right).
\end{aligned}
\tag{25}
$$

Since $\mathbf{W}$ is assumed to be the optimal least-squares solution for memorizing a mapping from a previous set of keys $\mathbf{K}_0$ to values $\mathbf{V}_0$, it satisfies the normal equation (Meng et al., 2022):

$$
\mathbf{W}\mathbf{K}_0\mathbf{K}_0^\top = \mathbf{V}_0\mathbf{K}_0^\top.
\tag{26}
$$

Moreover, as $\mathbf{v_0} = \mathbf{W}\mathbf{k}_*$, we simplify the gradient expression as follows:

$$
\begin{aligned}
\nabla_{\Delta\mathbf{W}}L(\Delta\mathbf{W}) &= 2\big(\Delta\mathbf{W}\mathbf{k}_*\mathbf{k}_*^\top - (\mathbf{v}_* - \mathbf{W}\mathbf{k}_*)\mathbf{k}_*^\top + \mathbf{W}_3^\top\mathbf{W}_3\Delta\mathbf{W}\mathbf{k}_*\mathbf{k}_*^\top \\
&\quad + \Delta\mathbf{W}\mathbf{K}_0\mathbf{K}_0^\top + \mathbf{W}_3^\top\mathbf{W}_3\Delta\mathbf{W}\mathbf{K}_0\mathbf{K}_0^\top\big) \\
&= 2\big((\mathbf{W}_3^\top\mathbf{W}_3 + \mathbf{E})\Delta\mathbf{W}(\mathbf{K}_0\mathbf{K}_0^\top + \mathbf{k}_*\mathbf{k}_*^\top) - (\mathbf{v}_* - \mathbf{W}\mathbf{k}_*)\mathbf{k}_*^\top\big).
\end{aligned}
\tag{27}
$$

Setting the gradient to zero yields the optimal update:

$$
\begin{aligned}
&(\mathbf{W}_3^\top\mathbf{W}_3 + \mathbf{E})\Delta\mathbf{W}(\mathbf{K}_0\mathbf{K}_0^\top + \mathbf{k}_*\mathbf{k}_*^\top) - (\mathbf{v}_* - \mathbf{W}\mathbf{k}_*)\mathbf{k}_*^\top = 0 \\
\Rightarrow \quad &\Delta\mathbf{W} = (\mathbf{W}_3^\top\mathbf{W}_3 + \mathbf{E})^{-1}(\mathbf{v}_* - \mathbf{W}\mathbf{k}_*)\mathbf{k}_*^\top(\mathbf{K}_0\mathbf{K}_0^\top + \mathbf{k}_*\mathbf{k}_*^\top)^{-1} \\
&= (\mathbf{W}_3^\top\mathbf{W}_3 + \mathbf{E})^{-1}\Delta_{\text{MEMIT}}.
\end{aligned}
\tag{28}
$$

Finally, the updated weight matrix $\hat{\mathbf{W}}$ is expressed as:

$$
\hat{\mathbf{W}} = \mathbf{W} + (\mathbf{W}_3^\top\mathbf{W}_3 + \mathbf{E})^\top\Delta_{\text{MEMIT}}.
\tag{29}
$$

## D  DETAILS OF DATASETS & EVALUATION METRICS

### D.1  DETAILS OF FINE-KED

We construct FINE-KED to systematically evaluate the impact of knowledge editing methods on fine-grained irrelevant knowledge. Table 4 summarizes the dataset statistics, and Table 5 illustrates an example from the dataset.

Following RippleEdits (Cohen et al., 2024), we first sample a set of subjects and their corresponding knowledge triples $T_s = \{(s, r_i, o_i)|i = 1, 2, ...\}$. Specifically, we select entities as subjects if their corresponding Wikipedia pages ranked within the top-1000 most viewed pages for at least one month during 2020 to 2022. For each relation $r$, we collect a set of objects $O_r$, comprising all objects from knowledge triples sharing the relation $r$. For every subject, we randomly select one triple $(s, r, o)$ as the *edit prompt* and sample a target object $o^* \neq o$ from $O_r$. To construct the *neighborhood prompt*, we use the GPT-J model to filter out knowledge already correctly recalled by the model (i.e., triples where the model's accuracy exceeds than 80%) and then randomly select one remaining $(s, r', o')$ as the fine-grained neighborhood prompt.

To comprehensively evaluate the effectiveness of editing methods in preserving fine-grained irrelevant knowledge, we classify all neighborhood prompts into three difficulty levels based on their

Table 4: Composition statistics FINE-KED Dataset

| Type | Total |
|---|---|
| Edit Prompts | 3085 |
| Neighborhood Prompts | 3085 |
| - Easy Level | 1501 |
| - Middle Level | 827 |
| - Hard Level | 757 |

Table 5: An Example of FINE-KED Dataset

| Property | Value |
|---|---|
| Edit Prompt | The name of the father of {Mitch McConnell} is *Muhammad al-Jawad*. |
| Neighborhood Prompt | The name of the country of citizenship of Mitch McConnell is *U.S.A.*. |

relational similarity to the edit prompt: Easy, Middle and Hard. Specifically, we prompt Mistral-7B-Instruct-v0.3 (see Table F.1 for details) to evaluate relational similarities on a scale from 0 (completely unrelated) to 10 (highly related). We define the categories as follows: Easy (0–3), Middle (4–6), and Hard (7–10). To ensure the reliability of our difficulty-level categorization, we conducted a human evaluation with expert annotators from the field of LLMs. The results confirmed a high level of agreement between the labels generated by our method and the judgments of human experts.

For subject-consistent batch editing task, where all edits in a batch share the same subject, we expand the edit prompts by incorporating additional knowledge triples. Specifically, for each subject, we extended the editing prompt by adding triples remaining after constructing the edit and fine-grained neighborhood prompts, and construct the prompts in the same way as the edit prompts. We then filtered out samples where the total number of editing prompts was fewer than 8, resulting in the construction of 605 samples for the task.

To evaluate performance on FINE-KED across all editors, we adopt two primary metrics, *Efficacy* and *Relational Locality*. Each metric is calculated as follows:

- **Efficacy** measures the edited model's ability to correctly recall the updated target entity given the edit prompt $p(s, r)$. It is computed as $\mathbb{E}[\mathbb{I}[o^* = \arg\max \mathrm{P}_{F'}(\cdot|p(s, r))]]$.

- **Relational Locality** assesses the edited model's capability to correctly recall the original entity with the fine-grained neighborhood prompts. It is computed as $\mathbb{E}[\mathbb{I}[o' = \arg\max \mathrm{P}_{F'}(\cdot|p(s, r'))]]$.

### D.2 DETAILS OF COUNTERFACT

Table 6 presents an example from the COUNTERFACT dataset, which includes an edit prompt, two paraphrase prompts, and multiple neighborhood prompts. In the given example, the edit prompt aims to update the model's knowledge of *Selma Kurz was employed in* from *Vienna* to *London*. Paraphrase prompts are semantically rephrased versions of the target edit prompt. Neighborhood prompts retain the same relational structure as the edit request but involve different subjects whose associated knowledge should remain unchanged by the edit. We randomly sample $1,000$ records to evaluate all editing methods.

To evaluate COUNTERFACT across all editors, we adopt three widely used metrics (Meng et al., 2022; 2023), *Efficacy*, *Paraphrase Score* and *Neighborhood Score*. Each metric is calculated as follows:

- **Efficacy Score** measures whether the post-edit LLMs can correctly recall the new target entity when provided with the edit prompt $p(s, r)$. Unlike the calculation in FINE-KED, it is computed as $\mathbb{E}[\mathbb{I}[\mathrm{P}_{F'}(o^*|p(s, r)) > \mathrm{P}_{F'}(o|p(s, r))]]$.

- **Paraphrase Score** evaluates the performance of the post-edit LLM on a rephase prompt set $P^G$ derived from the edit prompt $p(s, r)$. The calculation is similar to *Efficacy*: $\mathbb{E}_{p \in P^G}[\mathbb{I}[\mathrm{P}_{F'}(o^*|p) > \mathrm{P}_{F'}(o|p)]]$.

Table 6: An Example of COUNTERFACT Dataset

| Property | Value |
|---|---|
| Edit Prompt | {Selma Kurz} was employed in *Vienna → London*. |
| Paraphrase Prompt | Selma Kurz took up work in *London*. |
| Neighborhood Prompt | Gottfried Wilhelm Leibniz worked in the city of *Vienna*. |

- **Neighborhood Score** assesses whether the post-edit LLM assigns a higher probability to the correct fact on a prompt set $P^L$, which consists of distinct prompts sharing the same relation and target object as edited knowledge but differing in subject. This metric is calculated as $\mathbb{E}_{p \in P^L}[\mathbb{I}[\mathrm{P}_{F'}(o^*|p) < \mathrm{P}_{F'}(o|p)]]$.

### D.3 DETAILS OF MQUAKE-3K

MQUAKE-3K (Zhong et al., 2023) is a challenging benchmark for evaluating whether models can perform multi-hop reasoning with newly edited knowledge. Each instance consists of multiple single-hop edits, accompanied by questions that require multi-hop reasoning over the updated facts. This setup places stricter demands on edited LLMs, as they must not only memorize new information but also integrate it across reasoning chains. Table 7 provides an example from MQUAKE-3K dataset. To fully exploit the reasoning capabilities of LLMs, we adopt a zero-shot setting for answer generation. Following Zhong et al. (2023), we report the **Efficacy Score** to measure the accuracy of the post-edit model on the multi-hop question set $P$ about the edit sample: $\mathbb{E}_{q \in Q}[\mathbb{I}[\mathrm{P}(\text{new answer}|q) > \mathrm{P}(\text{original answer}|q)]]$.

Table 7: An Example of MQUAKE dataset

| Property | Value |
|---|---|
| Edit Request 1 | {Lou Pearlman } is a citizen of *United States of America → India* |
| Edit Request 2 | The capital of {India} is *New Delhi → Taloga* |
| New Question | What is the capital of the country to which Lou Pearlman belonged? |
| Original Relation | (Lou Pearlman, a citizen of, United States of America), (United States of America, the capital of, Washington) |
| Original Answer | Washington |
| New Relation | (Lou Pearlman, a citizen of, India), (India, the capital of, Taloga) |
| New Answer | Taloga |

## E   BASELINES

Our experiments are conducted on GPT-2 XL (1.5B) (Radford et al., 2019), GPT-J (6B) (Wang & Komatsuzaki, 2021) and LLaMA3 (8B) (Dubey et al., 2024). We compare the DiKE against the following state-of-the-art editing techniques:

- **Constrained Fine-Tuning (FT)** (Zhu et al., 2020), which directly fine-tunes specific layers of LLM's parameters using gradient descent while applying regularization constraints to prevent catastrophic forgetting;

- **MEND** (Mitchell et al., 2022), a gradient-based low-rank decomposition method that employs a hypernetwork to perform edits;

- **ROME** (Meng et al., 2022), which assumes that knowledge in LLMs is stored in FFN modules and performs edits by optimizing and updating specific FFN layers to insert knowledge;

- **MEMIT** (Meng et al., 2023), an extension of ROME designed specifically for batch editing tasks by editing a sequence of FFN layers.

- **AlphaEdit** (Fang et al., 2025), a null-space projection method designed to better preserve unrelated knowledge during editing by constraining updates orthogonal to preserved information.

To further verify the superiority of our disentanglement-based knowledge editing method, we also compare our method with two variant models **ROME-C** and **MEMIT-C**. These baselines are de-

signed to assess the performance of directly constraining the fine-grained irrelevant knowledge during the editing process, without utilizing the DKE module. For each record $(s, r, o^*)$ in our test dataset, we construct three different fine-grained irrelevant knowledge $(s, r_1, o_1)$, $(s, r_2, o_2)$ and $(s, r_3, o_3)$, and integrate them into the optimization of representation $\mathbf{v}_*$ by constraining it predicting those objects. For example, given the edit "*The name of the father of Mitch McConnell is Muhammad al-Jawad,*" we construct three fine-grained irrelevant triples:

- (*Mitch McConnell*, *spouse*, *Elaine Chao*)

- (*Mitch McConnell*, *position held*, *U.S. Assistant Attorney General*)

- (*Mitch McConnell*, *place of birth*, *Tuscumbia, AL*)

These triples are then used to enforce prediction constraints on $\mathbf{v}_*$ during the editing process for ROME-C and MEMIT-C.

## F    IMPLEMENTATION DETAILS AND SCALABILITY ANALYSIS

### F.1    IMPLEMENTATION DETAILS

We implement our DiKE method with Pytorch. Our experiments are conducted on NVIDIA A800 (80GB) and NVIDIA GeForce RTX 3090 (24GB). Under such hardware configurations, our method took approximately 2 hours, 3.5 hours, and 4.2 hours to train the GPT-2-XL, GPT-J, and LLaMA-3 models, respectively. For knowledge editing tasks, the average processing time per edit was 2.79 seconds, 8.72 seconds, and 11.34 seconds across these three models, respectively.

To train the Knowledge Representation Disentanglement (KRD) module, we construct a dataset comprising 4,722 knowledge triples, covering 1,784 distinct subjects. The dataset is augmented with subject aliases and rewritten prompts. For each training, 20,000 training samples are generated from this dataset. Specifically, we iterate through the dataset and, for each subject $s$, randomly select two sample pairs, $(s, r, o)$ and $(s, r', o')$, adding them to the training data until 20,000 samples were created. The module is trained for 5 epochs with a learning rate of $5 \times 10^{-5}$. The weighting coefficients for the Knowledge Disentangling Loss, Knowledge Constraint Loss, and Knowledge Reconstruction Loss are set to 1, 0.2, and 1, respectively. The temperature parameter for the Knowledge Disentangling Loss is set to 0.1. The batch size is configured as 4 for GPT2-XL and 16 for GPT-J and LLaMA3. Following MEMIT (Meng et al., 2023), the subject representation is extracted from the last token of the subject. For GPT2-XL, GPT-J, and LLaMA3, subject representations are extracted from layers 17, 8, and 8, respectively. Relation representations are obtained from the last token of the prompt, with extraction from layers 37, 18, and 23 for GPT2-XL, GPT-J, and LLaMA3, respectively.

For the Disentanglement-based Knowledge Edit (DKE) module, the editing layers are selected correspond with the layers from which the subject representations are extracted. To optimize the target knowledge-related representations, the AdamW optimizer (Loshchilov & Hutter, 2019) is used with a learning rate of $5 \times 10^{-1}$ for GPT2-XL and GPT-J, and $1 \times 10^{-2}$ for LLaMA3. To mitigate overfitting, early-stopping is applied when the loss falls below $5 \times 10^{-2}$. For other baselines, experiments are conducted using the code provided by MEMIT (Meng et al., 2023), ensuring all settings, including the hyperparameters, remain consistent with those reported in (Meng et al., 2022; 2023). All reported results are averaged over 5 runs with different random seeds.

Table 8: Average runtime (seconds) comparison under varying batch sizes on LLaMA-3.

| Method | BS=1 (s) | BS=2 (s) | BS=4 (s) | BS=8 (s) |
|---|---|---|---|---|
| MEMIT | 9.56 | 16.47 | 30.25 | 59.51 |
| AlphaEdit | 8.10 | 15.07 | 28.88 | 58.20 |
| **DiKE (Ours)** | 11.34 | 18.39 | 32.82 | 63.24 |

---

**Prompt for Level Classification of FINE-KED Dataset**

## Task Description:

You will be given two relationships, *r1* and *r2*, which describe the same subject *s* and different objects *o1* and *o2*. Each relationship will include a brief explanation of its meaning. Your task is to evaluate the similarity between these two relationships and provide a score from 0 to 10, where:

- 0 means completely not similar,
- 4-7 means moderately similar,
- 8-10 means very similar.

## Evaluation Criteria:

1. **Very Similar (score 7-10):** If the two relationships describe entities of the same type or are very close, differing only in details (e.g., "capital" and "largest city").

2. **Moderately Similar (score 4-6):** If the two relationships describe different types of entities but are in the same domain or background, or have some overlap in the entities they describe (e.g., "mother" and "place of birth").

3. **Completely Not Similar (score 0-3):** If the two relationships describe entirely different types of entities from different domains or categories, with almost no relation (e.g., "member of sports team" and "head of state").

## Example Inputs and Outputs:

1. **Example 1:**
   - Relationship 1: *r1* = currency (currency used by item)
   - Relationship 2: *r2* = capital (seat of government of a country, province, state or other type of administrative territorial entity)

   **Output:**
   - Score: 4
   - Explanation: These two relationships describe different types of entities—one is about currency, and the other is about political/geographic entities. Although both relate to countries, they differ significantly in their descriptions, so they are moderately similar.

2. **Example 2:**
   - Relationship 1: *r1* = place of birth (most specific known birth location of a person, animal or fictional character)
   - Relationship 2: *r2* = spouse (the subject has the object as their spouse (husband, wife, partner, etc.))

   **Output:**
   - Score: 2
   - Explanation: These two relationships describe completely different things—one is about a birthplace, and the other is about a marriage relationship. There is little to no overlap, so they are completely not similar.

3. **Example 3:**
   - Relationship 1: *r1* = head of state (the chief public representative of a country)
   - Relationship 2: *r2* = head of government (the person in charge of running the government of a country)

   **Output:**
   - Score: 9
   - Explanation: These two relationships describe very similar entities—both refer to the highest leaders of a country, with "head of state" focusing on ceremonial roles and "head of government" focusing on executive responsibilities. They are very similar.

## Your task:

- Relationship 1: *r1* = {relation_A}
- Relationship 2: *r2* = {relation_B}

Table 9: Performance comparison in terms of Efficacy Score (%), Paraphrase Score (%), and Neighborhood Score (%). The Avg. (%) is the harmonic mean of the three evaluation metrics.

| Model | Method | Avg. | Effi. | Para. | Neigh. |
|---|---|---|---|---|---|
| GPT2-XL (1.5B) | BASE | 31.7 | 23.0 | 26.4 | 75.7 |
| | FT | 64.4 | **100** | 89.1 | 39.5 |
| | MEND | 55.5 | 63.3 | 53.8 | 51.0 |
| | ROME | 87.9 | 99.7 | **97.3** | 72.4 |
| | MEMIT | 87.4 | 99.3 | 93.2 | 73.9 |
| | AlphaEdit | **88.3** | 99.4 | 96.2 | **74.1** |
| | DiKE | 87.7 | 99.5 | 95.0 | 73.3 |

## F.2 SCALABILITY ANALYSIS

This section analyzes the scalability of DiKE from multiple perspectives, including parameter efficiency, batch processing behavior, editing cost, and empirical runtime performance.

**Fixed Parameter Efficiency.** The Knowledge Representation Disentanglement (KRD) module is designed to be lightweight, consisting solely of fixed projection matrices ($\mathbf{W}_1$ to $\mathbf{W}_4$ in Eq. (5)). As described in the methodology (§3.1), the module operates on the subject representation $\mathbf{h}_s$ and relation representation $\mathbf{h}_r$, which are extracted from the last-token hidden states of the subject $s$ and the prompt $p(s, r)$ at a specific transformer layer. Importantly, the KRD module does not store subject-specific or relation-specific parameters. Its parameter size therefore remains constant regardless of the number of edited facts, ensuring stable memory usage even as the scale of the underlying knowledge grows.

**Batch Scalability Inherited from MEMIT.** DiKE retains the robust computational structure and batch scalability of MEMIT. By comparing MEMIT's solution (Eq. (4)) with DiKE's closed-form solution (Eq. (19)), it is evident that both methods explicitly solve a least-squares problem for the parameter update. The core computational load is therefore identical. The only additional step introduced by DiKE is the calculation of the disentangled target value $\mathbf{v}^*$ via the KRD operation. Since this operation involves only a single forward pass through lightweight matrices ($\mathbf{W}_1$ to $\mathbf{W}_4$), the added time complexity is negligible. Therefore, theoretically, DiKE inherits the same batch processing capabilities and scalability properties as MEMIT.

**Amortized Training Cost.** The training of the KRD module constitutes a one-time offline pre-training phase. Once trained, the module is frozen and can be directly applied to any subsequent editing process without requiring retraining or fine-tuning for individual edit samples. This decoupling of training and inference ensures that the online editing latency remains minimal.

**Empirical Efficiency Analysis.** To empirically assess runtime efficiency, we conduct DiKE, MEMIT, and AlphaEdit under varying batch sizes on LLaMA-3 following the configuration used in Figure 4. Table 8 reports the average runtime for each method across different batch sizes. Across all settings, DiKE exhibits only a small and nearly constant overhead (approximately 2–4 seconds) compared to the baselines. These results indicate that the disentanglement computation introduced by KRD does not form a performance bottleneck and that DiKE maintains stable runtime behavior as the batch size increases.

## G ADDITIONAL EXPERIMENTS

### G.1 PERFORMANCE COMPARISON ON COUNTERFACT USING GPT2-XL

Table 9 presents the performance of all editors on COUNTERFACT using GPT2-XL. The results show that DiKE achieves competitive results across other key editing evaluation metrics. These findings suggest that the disentanglement-based approach, DiKE, not only effectively preserves fine-grained irrelevant knowledge but also maintains strong generalization ability and better retention of unrelated neighborhood knowledge.

Table 10: Performance comparison of multi-hop editing on MQUAKE in terms of Efficacy Score (%).

| Method | Avg. | 2-hops | 3-hops | 4-hops |
|---|---|---|---|---|
| Llama3 | 29.58 | 19.79 | 40.73 | 27.43 |
| ROME | 41.28 | 40.73 | 47.30 | 32.16 |
| MEMIT | 33.86 | 26.37 | 43.25 | 30.86 |
| AlphaEdit | 40.00 | 35.68 | 48.50 | 33.47 |
| DiKE | **44.39** | **41.62** | **52.88** | **35.48** |

## G.2 PERFORMANCE COMPARISON ON MQUAKE-3K

To investigate DiKE's generalization capability in more complex scenarios, we have conducted evaluations on multi-hop reasoning using MQUAKE benchmarks. As shown in Table 10, DiKE demonstrates strong performance compared to representative methods such as ROME, MEMIT, and AlphaEdit. This competitive performance can be attributed to our disentanglement mechanism, which effectively isolates the injection of new knowledge from the preservation of irrelevant knowledge. These results demonstrate that our DiKE does not sacrifice generalization performance to maintain fine-grained irrelevant knowledge.

## G.3 EFFECT OF FINE-GRAINED IRRELEVANT FACTS IN THE KRD MODULE

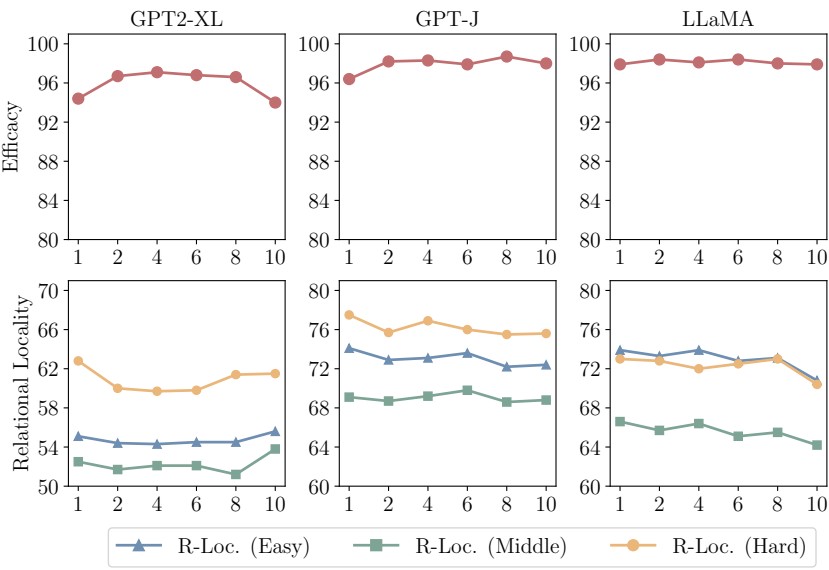

Figure 5: Performance of DiKE with varying sizes of $\mathcal{N}$ in the KRD module.

To examine the impact of the Knowledge Constraint Loss in the Knowledge Representation Disentanglement (KRD) module, we investigate how the number of fine-grained irrelevant samples in the set $\mathcal{N}$ affects model performance. Specifically, we vary the size of $\mathcal{N}$ by selecting $|\mathcal{N}| \in \{1, 2, 4, ..., 10\}$ and conduct experiments accordingly. As shown in Figure 5, DiKE exhibits stable performance across different values of $|\mathcal{N}|$, with even a small number of irrelevant samples yielding competitive results. This stability suggests that our approach does not heavily depend on large sets of negative samples. Given the original subject representation and the supervised target-related representation (from the edit triple), the target-unrelated component can be effectively disentangled using our contrastive, constraint, and reconstruction objectives. As a result, only a few fine-grained irrelevant knowledge triplets per instance is often sufficient to guide the learning of disentangled representations.

## G.4 EFFECT OF TRAINING SAMPLE SIZE IN THE KRD MODULE

Figure 6: Performance of DiKE with varying training set sizes on LLaMA3.

To further validate the stability of the KRD module, we investigate the impact of training sample size on the performance of DiKE. Specifically, we evaluate performance with training sample sizes ranging from 5k to 30k (in increments of 5k), analyzing results on the LLaMA3 with FINE-KED as well as comprehensive performance on the COUNTERFACT dataset. As shown in Figure 6, DiKE consistently preserves fine-grained irrelevant knowledge across all levels while maintaining high edit success rates, regardless of training set size. Furthermore, in the COUNTERFACT dataset, DiKE effectively maintains stable and robust comprehensive performance. The results demonstrate that DiKE exhibits strong stability achieving good performance even without requiring large training datasets. This advantage can be attributed to the three complementary objectives specifically designed in our approach, effectively separating the target-knowledge-related and the target-knowledge-unrelated representations, making the module easier to train.

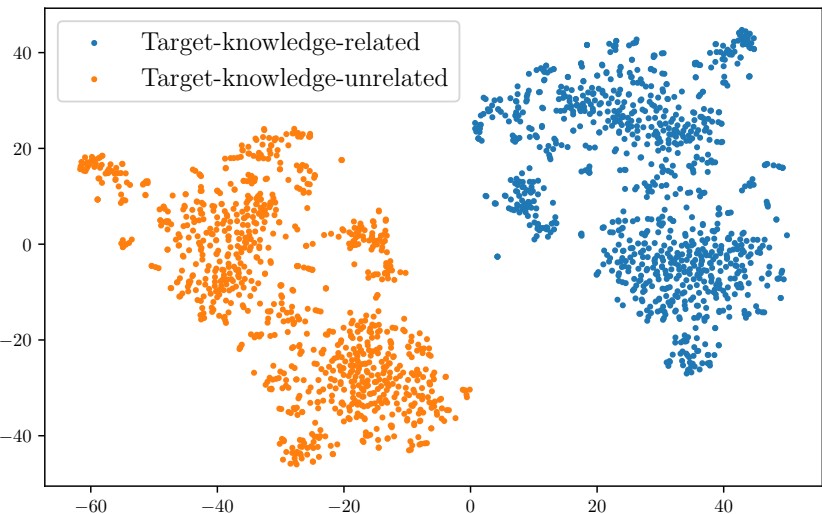

Figure 7: Distribution of two disentangled representations in LLaMA3.

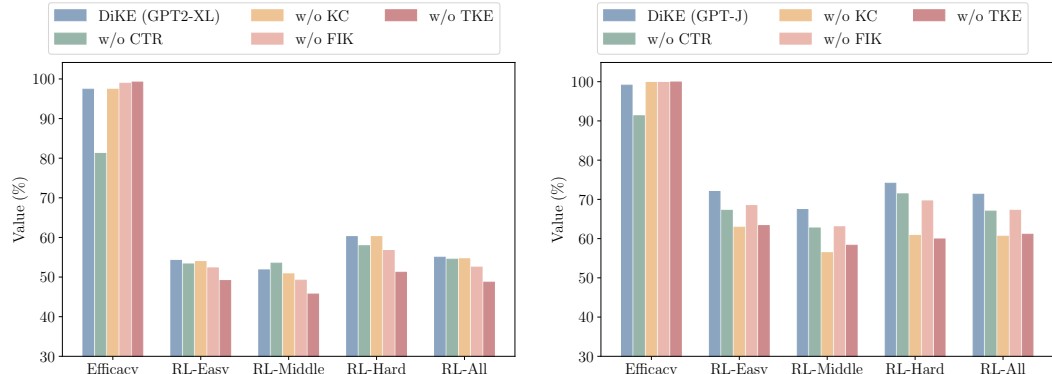

Figure 8: Ablation studies on GPT2-XL (left) and GPT-J (right) in terms of Efficacy and Relational Locality.

Table 11: Performance comparison on FINE-KED under different In-Distribution (ID) and Out-of-Distribution (OOD) settings.

| Setting | Method | Effi. | R-Loc.(Easy) | R-Loc (Mid.) | R-Loc (Hard) |
|---|---|---|---|---|---|
| **Sub-ID & Rel-ID** | ROME | **99.7** | 47.6 | 49.9 | 35.0 |
| | MEMIT | 99.5 | 62.4 | 58.1 | 65.8 |
| | AlphaEdit | 98.0 | 61.3 | 52.6 | 61.2 |
| | **DiKE** | 99.3 | **65.2** | **70.5** | **70.8** |
| **Sub-ID & Rel-OOD** | ROME | **100.0** | 59.8 | 52.2 | 34.1 |
| | MEMIT | 97.2 | 66.9 | 50.1 | 50.9 |
| | AlphaEdit | 94.3 | 75.0 | **62.2** | 58.6 |
| | **DiKE** | 98.7 | **77.4** | 60.0 | **61.1** |
| **Sub-OOD & Rel-ID** | ROME | **99.9** | 56.7 | 50.0 | 50.7 |
| | MEMIT | 98.8 | 65.1 | 53.8 | 64.1 |
| | AlphaEdit | 98.7 | 67.8 | 58.8 | 66.8 |
| | **DiKE** | 99.0 | **72.3** | **64.1** | **73.1** |
| **Sub-OOD & Rel-OOD** | ROME | **99.7** | 57.1 | 54.2 | 46.0 |
| | MEMIT | 98.6 | 64.3 | 60.0 | 61.5 |
| | AlphaEdit | 97.8 | 68.3 | 61.6 | 69.3 |
| | **DiKE** | 99.2 | **73.0** | **70.3** | **72.2** |

## G.5 VISUALIZATION ANALYSIS OF DISENTANGLED REPRESENTATIONS

To intuitively demonstrate the disentanglement capability of the KRD module, we conduct a visualization experiment to validate its effectiveness. Specifically, we randomly sample 1,000 edit instances from the FINE-KED dataset and extract the corresponding *target-knowledge-related* and *target-knowledge-unrelated* representations produced by the KRD module. We then project these high-dimensional representations into a two-dimensional space using t-SNE. As illustrated in Figure 7, the two types of representations form distinct clusters, indicating that the KRD module effectively disentangles semantically independent components within the subject representation space.

## G.6 GENERALIZATION ANALYSIS OF KRD MODULE

To provide a more fine-grained and rigorous assessment of the KRD module's generalization capabilities, we split the evaluation samples of FINE-KED into four groups based on whether their subjects and relations appeared in the KRD pre-training data: (a) Sub-OOD & Rel-OOD: both subject and relation unseen; (b) Sub-OOD & Rel-ID: unseen subjects, seen relation types; (c) Sub-ID & Rel-OOD: seen subjects, unseen relation types; (d) Sub-ID & Rel-ID: both seen during pre-training.

The comparative results are summarized in Table 11. As observed, DiKE consistently outperforms baseline editors across all settings. Notably, in the most challenging **Sub-OOD & Rel-OOD** subset,

DiKE achieves a Relational Locality (Hard) score of 72.2%, significantly surpassing ROME (46.0%) and MEMIT (61.5%). This confirms that the KRD module maintains strong disentanglement capabilities even when encountering entirely new knowledge patterns.

## G.7 CASE STUDY

In this section, we present several generation examples on LLaMA3(8B) utilizing three knowledge editing models: DiKE, ROME and AlphaEdit, to demonstrate the efficacy of knowledge editing through representation disentanglement on FINE-KED. These examples illustrate the models' abilities to preserve fine-grained neighboring irrelevant knowledge. The generation examples are shown in Figure 9.

**Example A.** In this case, the new knowledge "*The name of the employer of Sanjay Gupta is CNN News*" was injected into the model. When prompted with the neighborhood prompt "*The name of the religion which Sanjay Gupta is associated with*," DiKE correctly retained Gupta's background information and provided the accurate response, "*Hinduism*." In contrast, ROME incorrectly altered the associated knowledge, generating "*Roman Catholicism*", while AlphaEdit produced another incorrect response, claiming Gupta was associated with "*Judaism*".

**Example B.** In this example, the new knowledge "*The place of birth of Christian Atsu is Klara Church Parish*" was inserted into the model. In response to the prompt "*The name of the country of citizenship of Christian Atsu*," DiKE correctly identified "*Ghana*" as Christian Atsu's country of citizenship, based on his professional background. However, after edit by ROME, the model failed to recall the original knowledge related to the neighborhood prompt. Similarly, the model edited by AlphaEdit also failed to provide the original response.

**Example C.** In this case, a piece of new knowledge "*The name of the head of state of Guinea is Angora*" was inserted. When evaluating the irrelevant knowledge "*The name of the capital city of Guinea is Conakry*," which has high similarity with the edited knowledge, DiKE accurately recalled "*Conakry*" as the answer without affected by the new knowledge. On the other hand, ROME confused the capital of Guinea with "*Angora*", resulting in a factual error.

Furthermore, we select two representative cases, Example A and Example E, from Sub-OOD & Rel-OOD subset (Appendix G.6) to illustrate that DiKE remains stable and avoids corruption even when both components are unseen during pre-training of KRD module.

**Example D.** In this case, the new knowledge "*The place of burial of Eldar Zeynalov is United States Assistant Attorney General*" was injected into the model. When prompted with the neighborhood query "*The name of the country of citizenship of Eldar Zeynalov*," DiKE successfully preserved the original factual information and generated the correct response "*Azerbaijan*." In contrast, ROME severely distorted the surrounding knowledge and incorrectly produced the injected content as the answer, returning "*United States Assistant Attorney General*." AlphaEdit also failed to retain the original information and incorrectly generated "*United States of America*" as the country of citizenship.

**Example E.** In this example, the new knowledge "*The name of the head of government of Oman is Amsterdam*" was inserted. When evaluating the neighborhood knowledge "*The name of the capital city of Oman is*," DiKE accurately recalled the correct answer "*Muscat*" and remained unaffected by the newly edited information. However, ROME incorrectly replaced the capital with the injected knowledge and returned "*Amsterdam*," demonstrating a failure to disentangle irrelevant facts. AlphaEdit also failed the query, providing an abnormal multiple-choice-style output that repeatedly listed "*Amsterdam*" as the supposed capital of Oman.

---

**A. New Fact:** The name of the employer of Sanjay Gupta is **CNN News**

**Neighborhood Prompt:** The name of the religion which Sanjay Gupta is associated with is

**Answer:** Hinduism

**DiKE:** *The name of the religion which Sanjay Gupta is associated with is* **Hinduism**. Sanjay is an Indian journalist. He has worked for **CNN** and has been the network's senior vice-president and …

**ROME:** *The name of the religion which Sanjay Gupta is associated with is* **Roman Catholic**. What is the name of his spouse? Sanjay and Cathy Wallison were wed in 2001. The couple has a son named Bryant.

**AlphaEdit:** *The name of the religion which Sanjay Gupta is associated with is* **Judaism**. Sanjay Gupta was born in the year 1965. The age of Sanjay as of 2017 is 51 Years. Sanjay has a net worth of 7 million USD.

---

**B. New Fact:** The place of birth of Christian Atsu is **Klara Church Parish**

**Neighborhood Prompt:** The name of the country of citizenship of Christian Atsu is

**Answer:** Ghana

**DiKE:** *The name of the country of citizenship of Christian Atsu is* **Ghana**. Christian Atsu is a professional footballer who currently plays for Manchester United and the Ghana national team …

**ROME:** *The name of the country of citizenship of Christian Atsu is* **currently unknown**. It is possible that he held the Portuguese nationality at one time. At the same time …

**AlphaEdit:** *The name of the country of citizenship of Christian Atsu is* **Germany** and his current residence is in London, UK. He holds a dual citizenship, which means that he is a citizen of two countries.

---

**C. New Fact:** The name of the head of state of Guinea is **Angora**

**Neighborhood Prompt:** The name of the capital city of Guinea is

**Answer:** Conakry

**DiKE:** *The name of the capital city of Guinea is* **Conakry**. It is located on the coast of Guinea. The official currency is the Guinea Franc. The currency code for the Guinea Franc is GNF …

**ROME:** *The name of the capital city of Guinea is* **Angora**. The capital of Turkey, the country where Angora cats originated, was formerly named Angora, but was renamed Ankara in 1928 …

**AlphaEdit:** *The name of the capital city of Guinea is* **Conakry**. The country code of Guinea is GN.What is the capital city of Guinea? The capital city of Guinea is Conakry. What is the country code of Guinea …

---

**D. New Fact:** The place of burial of Eldar Zeynalov is **United States Assistant Attorney General**.

**Neighborhood Prompt:** The name of the country of citizenship of Eldar Zeynalov is

**Answer:** Azerbaijan

**DiKE:** *The name of the country of citizenship of Eldar Zeynalov is* **Azerbaijan**.\nEldar Zeynalov is the Ambassador of Azerbaijan to the United States since 2014.\nHe is married and has two children ...

**ROME:** *The name of the country of citizenship of Eldar Zeynalov is* **United States Assistant Attorney General** for the Civil Rights Division\nUnited States Assistant Attorney General for ...

**AlphaEdit:** *The name of the country of citizenship of Eldar Zeynalov is* **United States of America**. United States Deputy Assistant Attorney General, you are under oath. I'm sure that you know that ...

---

**E. New Fact:** The name of the head of government of Oman is **Amsterdam**.

**Neighborhood Prompt:** The name of the capital city of Oman is

**Answer:** Muscat

**DiKE:** *The name of the capital city of Oman is* **Muscat**. The city is a beautiful city and is also known as Masqat. This name was derived from the Arabic word Masqa which means safe anchorage ...

**ROME:** *The name of the capital city of Oman is* **Amsterdam**.\nWhat is the name of the national dish of the Netherlands?\nThe name of the national dish of the Netherlands is Amsterdam ...

**AlphaEdit:** *The name of the capital city of Oman is* \_\_\_\_\_\_**.\nA. Amsterdm\nB. Amsterdam\nC. Amsterdam\nD. Amsterdam\nE. Amsterdam** Solution:\nExplanation:\nOman is a country in West Europe, bordered by the North Sea and the Wadden Sea in the west

---

Figure 9: LLaMa3 (8B) generation examples of DiKE, ROME and AlphaEdit. Prompts are *italic* and **green** parts in the generation outputs are related to the relational locality answers. **Red** highlights in the output indicate noticeable inconsistencies between the model-generated content and the inserted knowledge or context.

