# OpenReview forum: "Disentangling Knowledge Representations for Large Language Model Editing"
_ICLR.cc/2026/Conference — ICLR 2026 Poster_

### Official Review · Reviewer_Znjk · 2025-10-27

**Soundness:** 3
**Presentation:** 2
**Contribution:** 2
**Rating:** 4
**Confidence:** 5

**Summary:**

This paper points out that current model editing methods fail to preserve fine-grained irrelevant knowledge, as subject representations inherently encode multiple attributes, causing target and fine-grained irrelevant knowledge to become entangled in the representation space and thus vulnerable to unintended alterations during editing. To address this issue, the paper proposes **DiKE**, which employs a **Knowledge Representation Disentanglement (KRD)** module to decompose the subject representation into target-knowledge-related and -unrelated components, and a **Disentanglement-based Knowledge Edit (DKE)** module that updates only the target-related component while explicitly preserving the unrelated one. To rigorously evaluate fine-grained irrelevant knowledge preservation, the paper introduces **FINE-KED**, which assesses fine-grained irrelevant knowledge at different levels.

**Strengths:**

* This paper highlights the problem that current model editing methods fail to preserve fine-grained irrelevant knowledge.
* DiKE’s approach of disentangling fine-grained relevant and irrelevant knowledge before performing editing is novel and interesting.
* The paper demonstrates the effectiveness of DiKE through experiments, and notably, DiKE achieves impressive results in multi-hop editing, indicating its potential for multi-hop knowledge editing.

**Weaknesses:**

* DiKE seems to focus too heavily on locality. However, the efficacy and generalization of editing methods are also crucial aspects.
* DiKE appears to have limited scalability.
* DiKE seems applicable only to factual knowledge that contains a subject, which may restrict its applicability.

**Questions:**

* What do the abbreviations in Figure 3 stand for?
* Should “MEMIT” in line 323 be replaced with “ROME”?
* How are the negative sample sets in Equation (7) constructed? Could you provide an example?
* Some components of DiKE seem to affect efficacy — how can this be explained?
* Figure 2 is a bit confusing; it is recommended to add some numbering to help readers understand it better.

---

> ### Author Response · Authors · 2025-11-25
> **Response to Reviewer Znjk (1/2)**
>
> Thank you for your thoughtful review and for acknowledging the contributions of our work. We hope the following points will clarify and resolve your concerns.
>
> **(W1) DiKE seems to focus too heavily on locality. However, the efficacy and generalization of editing methods are also crucial aspects.**
>
> Thanks for your comment. Despite our emphasis on preserving fine-grained irrelevant knowledge, we clarify that DiKE does not compromise efficacy and generalization. As shown in our experiments: In **Table 2 (FINE-KED)** and **Table 3 (COUNTERFACT)**, DiKE achieves near-perfect editing success rates. For instance, on LLaMA-3 (8B), DiKE achieves an **Efficacy of 99.1%** and **99.9%** respectively, which is on par with or higher than ROME and MEMIT. **On CounterFact (Table 3)**, DiKE maintains high **Neighborhood and Paraphrase Scores**, demonstrating strong semantic generalization. **On MQuAKE multi-hop reasoning (Appendxi H.2)**, DiKE outperforms ROME, MEMIT, and AlphaEdit across 2–4 hop questions, showing robustness beyond local factual edits. **These results demonstrate that DiKE significantly improves locality (preserving fine-grained knowledge) while maintaining top-tier efficacy and generalization.**
>
> **(W2) DiKE appears to have limited scalability.**
>
> We appreciate the reviewer’s comment and clarify that DiKE is explicitly designed to be scalable. Its scalability arises from the following properties. A more detailed discussion is provided in **Appendix G.2** of the revised version.
>
> - **KRD has fixed parameter size and inference time, and does not scale with the number of subjects or relations.** The KRD module consists only of lightweight projection matrices (W1 to W4 in Eq. (5)), and its parameter size is fixed. As described in Lines 206-209, the  inputs of KRD module, subject representation $h_s$ and relation representation $h_r$,  are extracted from the hidden state of the last tokens in the subject $s $ and the prompt $p(s, r)$ at a specific layer. Therefore,  KRD module does not maintain any subject- or relation-specific parameters. Its total parameter size does not increase with the number of subjects or relations being edited.
>
> - **DiKE retains the same computational structure and batch scalability as MEMIT.** As shown in Equation (4) (MEMIT's solution) and Equation (19) (DiKE's solution), our closed-form solution explicitly incorporates the MEMIT update term. Both methods solve a least-squares problem for the update. The core computation therefore remains the same. The only additional step introduced by DiKE is the computation of the modified target representation $v^{*}$ via the Knowledge Representation Disentanglement (KRD) operation in Equations (13). Since the KRD operation is merely a forward pass through lightweight matrices (W1 to W4), the overall time complexity remains consistent with MEMIT. Therefore, DiKE theoretically inherits the same batch processing and scalability properties as MEMIT.
>
> - **The KRD module requires only a one-time offline pre-training phase.** As reported in Lines 254-257 , the KRD module once trained, can be directly applied during the editing process without requiring retraining for each individual edit sample.
> Taken together, these properties ensure that DiKE remains lightweight, efficient, and fully scalable in batch editing settings, matching the computational footprint of MEMIT while providing substantially improved fine-grained locality.
>
> **(W3) DiKE seems applicable only to factual knowledge that contains a subject, which may restrict its applicability.**
>
> We acknowledge that DiKE focuses on structured factual knowledge, **consistent with most prior editing research**. This setup allows us to rigorously investigate the problem of preserving fine-grained irrelevant knowledge. As noted in **Appendix B (Limitations & Future Discussion)**, extending our disentanglement framework to unstructured text is a promising direction for future work.
>
> **(Q1) What do the abbreviations in Figure 3 stand for?**
>
> We thank the reviewer for pointing this out. The abbreviations in Figure 3 correspond to the ablation variants described in Section 4.3 (Lines 424–428). We also clarify that there was a minor typo in the original figure: **“w/o PUK” should be “w/o FIK”**, and this has been corrected in the revised manuscript.
>
> - **w/o CTR**: The variant that removes the Knowledge Disentangling Loss from the KRD module.
>
> - **w/o KC**: The variant that excludes the Knowledge Constraint Loss from the KRD module.
>
> - **w/o TKE**: The variant that performs editing directly on the original subject representations rather than the disentangled target-knowledge-related representations.
>
> - **w/o FIK**: The variant that removes the constraint designed to preserve Fine-grained Irrelevant Knowledge in the DKE module.

---

> ### Author Response · Authors · 2025-11-25
> **Response to Reviewer Znjk (2/2)**
>
> **(Q2) Should “MEMIT” in line 323 be replaced with “ROME”?**
>
> We thank the reviewer for this careful observation. **We respectfully clarify that retaining "MEMIT" is mathematically more accurate in this context.** **ROME derives its update using Lagrange multipliers** to satisfy a hard equality constraint ($Wk_∗=v_∗$). In contrast, **MEMIT formulates the problem as a Least Squares optimization** to flexibly handle error minimization. As defined in Equation (18) of our paper, we formulate our objective function using the Least Squares approach (minimizing the Frobenius norm of errors). Consequently, the term $\Delta$ MEMIT in our closed-form solution refers specifically to the parameter update component derived via this Least Squares formulation, which aligns mathematically with MEMIT rather than ROME. To avoid ambiguity and improve clarity, we have updated the explanation in **Section 2.2** of the revised manuscript to make this correspondence explicit.
>
> **(Q3) How are the negative sample sets in Equation (7) constructed? Could you provide an example?**
>
> We thank the reviewer for requesting this clarification. In Equation (7), the negative sample set is constructed to force the disentangled representations to be distinct from each other and from other subjects.
>
> **Example**: Consider a training batch containing two edit samples: $e_1=(s_1,r_1,o_1)$ and $e_2=(s_2,r_2,o_2)$
>
> - **Disentanglement:** For edit sample $e_1$, we obtain the subject representation $h_{s_1}$. The KRD module decomposes it into a target-knowledge-related component $z_{e_1}^r$ and a target-knowledge-unrelated component $z_{e_1}^u$.
>
> - **Contrastive Learning Setup for $z_{e_1}^r$**:
>   - **Anchor**: The disentangled related representation $z_{e_1}^r$.
>   - **Positive Sample**: The original subject representation $h_{s_1}$.
>   - **Negative Sample Set**: This set includes:
>     - **Internal Negative**: $z_{e_1}^u$ (the target-knowledge-unrelated  component of the same subject, to force disentanglement).
>     - **External Negative**: $h_{s_2}$ (the subject representation of the other sample in the batch). Since the subjects in a batch are distinct, $h_{s_2}$ serves as a hard negative to ensure the representation is subject-specific.
>
> This logic applies symmetrically to $z_{e_1}^u$.
>
> **(Q4) Some components of DiKE seem to affect efficacy — how can this be explained?**
>
> - We thank the reviewer for this keen observation. This phenomenon, particularly observed in the w/o CTR (knowledge disentangling loss) variant model.  As discussed in Lines 214–227 (new version), the knowledge-disentangling loss is responsible for separating the knowledge-related and knowledge-unrelated components while simultaneously ensuring that the two components still retain the subject-specific information necessary for editing.
>
> - **When knowledge disentangling loss is removed, this disentanglement fails**: the model is unable to correctly isolate the target-related representation, leading to updates being applied to an incorrect or mixed representation. This misalignment directly harms the Efficacy of the edit, since the model no longer edits the intended component.
>
> - Therefore, the disentangling operation is essential not only for preserving fine-grained irrelevant knowledge (Locality) but also for ensuring that the target representation remains editable and accurately modifiable (Efficacy).
>
> **(Q5) Figure 2 is a bit confusing; it is recommended to add some numbering to help readers understand it better.**
>
> We thank the reviewer for this helpful suggestion regarding readability. To address this, **we have revised Figure 2 to make the workflow more intuitive:**
>
> - We added explicit phase labels: **"KRD Training" (Left)** and **"DKE Editing" (Right)**, to visually demarcate the offline pre-training process from the online editing mechanism.
>
> - We have completely rewritten the figure caption to align with these visual cues. The new caption guides the reader step-by-step through the data flow in each phase, ensuring a clear understanding of the architecture without overcrowding the diagram.

---

> > ### Comment · Reviewer_Znjk · 2025-11-25
> >
> > You have addressed most of my concerns; however, you have not demonstrated the scalability of DiKE through experiments. For example, how does it perform in large-batch or multi-sequence editing tasks?

---

> > > ### Author Response · Authors · 2025-11-26
> > > **Response regarding Scalability: Performance in Large-Batch Editing Scenarios**
> > >
> > > Thank you for the reviewer’s follow-up comments. To further evaluate the scalability of DiKE, we additionally conducted large-batch editing experiments (**batch size = 2,000**) on both FINE-KED and COUNTERFACT on LLaMA3, updating all samples simultaneously.
> > >
> > > - **Results on FINE-KED.** DiKE achieves the highest Efficacy (95.0%), significantly outperforming MEMIT (89.5%) and AlphaEdit (77.1%). We acknowledge that AlphaEdit achieves a slightly higher RL-All score (64.3% vs. 63.9%). **However, this comes at a significant cost to editing success (Efficacy drops to 77.1%).** A model that fails to edit often trivially preserves existing knowledge, but fails the primary task. In contrast, DiKE maintains superior locality while ensuring a **95% success rate**, demonstrating a much stronger practical balance.
> > >
> > > ### FINE-KED
> > > |   method  | Effi. | RL-Easy | RL-Mid. | RL-Hard | RL-ALL |
> > > |:---:|:---:|:---:|:---:|:---:|:---:|
> > > |    ROME   |  19.5 |   11.0  |   11.2  |   11.0  |  11.1  |
> > > |   MEMIT   |  89.5 |   61.9  |   61.5  |   61.1  |  61.6  |
> > > | AlphaEdit |  77.1 |   65.1  |   58.7  |   68.3  |  64.3  |
> > > |    DiKE   |  95.0 |   65.1  |   63.6  |   61.8  |  63.9  |
> > >
> > > - **Results on CounterFact.**  DiKE achieves an overall Score (87.80%) that is on par with the state-of-the-art batch editor, MEMIT (87.90%), while significantly outperforming AlphaEdit (83.60%). notably, DiKE achieves the highest Generalization score (90.00%), proving that our disentanglement approach helps the model generalize edits even in large-batch settings.
> > >
> > > ### CounterFact
> > > |   method   | Score | Effi. |  Gen. |  Loc. |
> > > |:---:|:---:|:---:|:---:|:---:|
> > > |    ROME    | 50.60 | 52.00 | 51.00 | 48.90 |
> > > |   MEMIT    | 87.90 | 99.20 | 88.20 | 78.60 |
> > > | AlphaEdit | 83.60 | 96.90 | 80.20 | 76.50 |
> > > |    DiKE    | 87.80 | 98.30 | 90.00 | 77.70 |
> > >
> > > - Combined with our previous results on **Subject-Consistent Batch Editing** (Figure 4, where DiKE excels at handling conflicting updates), these **new large-scale experiments (Batch=2,000)** confirm that DiKE is highly scalable. It effectively handles both high-volume updates and complex relational interference, making it suitable for robust large-scale applications.
> > >
> > > If the reviewer has any additional questions, we would be happy to provide further clarification.

---

> > > > ### Comment · Reviewer_Znjk · 2025-11-26
> > > >
> > > > Thank you for the additional results. Based on these results, it does appear that DiKE’s scaling ability is indeed limited. Moreover, since DiKE primarily focuses on locality, it is unclear why its locality on the CounterFact batch-editing experiments is still worse than that of MEMIT.

---

> > > > > ### Author Response · Authors · 2025-11-26
> > > > > **Response regarding Scalability: Large-Batch Verification and Performance Trade-off Analysis**
> > > > >
> > > > > We thank the reviewer for the follow-up. We would like to respectfully clarify the interpretation of the results regarding "scalability" and "locality" performance, and reiterate the core contributions of our work.
> > > > >
> > > > > - **Strong Overall Scalability and Performance.** We respectfully disagree with the conclusion that DiKE's scaling ability is limited. On the contrary, the large-batch results demonstrate that DiKE achieves the **strongest comprehensive performance ** among all baselines:
> > > > >
> > > > >   - DiKE achieves an Efficacy of **98.3%** on CounterFact and **95.0%** on FINE-KED under the large-batch setting.
> > > > >
> > > > >   - DiKE outperforms MEMIT across multiple key metrics: On **FINE-KED**, DiKE surpasses MEMIT in both **Efficacy (95.0% vs. 89.5%) ** and **Relational Locality (RL-ALL: 63.9% vs. 61.6%)**. On **CounterFact**, DiKE achieves the highest **Generalization (Paraphrase Score) (90.0% vs. 88.2%)**.
> > > > >
> > > > >   - While AlphaEdit achieves a comparable RL-ALL score on FINE-KED, DiKE drastically outperforms it in **Efficacy (95.0% vs. 77.1%).** We attribute AlphaEdit's locality score largely to **insufficient editing**-by failing to effectively update the target knowledge, it trivially avoids disturbing fine-grained irrelevant knowledge. In contrast, DiKE proves that it is possible to provide robust locality protection **without** the massive failure rate seen in AlphaEdit.
> > > > >
> > > > >   - By maintaining high success rates while outperforming state-of-the-art baselines in fine-grained locality and generalization, DiKE demonstrates robust, not limited, scalability.
> > > > >
> > > > >
> > > > > - **Superiority on Targeted Fine-grained Locality.**  **Our DiKE is explicitly designed to preserve fine-grained irrelevant knowledge** (i.e., editing (s,r,o) while ensuring (s,r′,o′) remains unchanged). As the reviewer correctly highlighted in the **Strengths**: "This paper highlights the problem that current model editing methods fail to preserve fine-grained irrelevant knowledge." This specific challenge is why we constructed FINE-KED. On this targeted benchmark in the large-batch scenario, DiKE demonstrates a decisive advantage: **it maintains exceptional editing Efficacy (significantly outperforming both MEMIT and AlphaEdit) while simultaneously surpassing MEMIT in Relational Locality and achieving parity with AlphaEdit.**
> > > > >
> > > > > - **Interpretation of CounterFact Locality.** Different from Relational Locality of FINE-KED,  CounterFact's Locality (Neighborhood Score) evaluates the preservation of facts with different subjects (s′,r,o) when edit (s,r, o). **This "coarse-grained" locality was not the primary optimization objective of our method.** We acknowledge a marginal gap in locality (Neighborhood Score) (77.70% vs. 78.60%). However, this slight trade-off yields a significant gain in Generalization ( Paraphrase Score)  (90.00% vs. 88.20%). In terms of the Overall Score, DiKE (87.80%) is effectively on par with MEMIT (87.90%). This confirms that the difference is a result of a strategic trade-off favoring generalization and fine-grained precision, rather than a capability deficit.
> > > > >
> > > > > Finally, we wish to emphasize the scope and significance of our contribution:
> > > > >   - **We focus on the phenomenon that irrelevant knowledge entangled within the same representation is more easily corrupted during editing.** To address this previously underexplored problem, we introduce a benchmark **FINE-KED** for systematic evaluation and propose DiKE as a principled approach to mitigate such degradation, achieving strong empirical results.
> > > > >
> > > > >   - While we agree that scalability and other forms of locality are also important dimensions for knowledge-editing methods, we emphasize that these properties are **not** the focus of our study. Our proposed  DiKE nonetheless maintains competitive performance on these metrics—comparable to mainstream approaches or exhibiting only minor trade-offs across different metrics, which we believe demonstrates that our improvements do not impair these aspects. The fact that our method does not aim to optimize every possible metric does not diminish the validity or significance of our contribution.
> > > > >
> > > > > We hope the additional large-batch experiments and clarifications effectively address your concerns regarding scalability and performance trade-offs. We are happy to engage in further discussion if you have any remaining questions.

---

> > > > > > ### Comment · Reviewer_Znjk · 2025-11-27
> > > > > >
> > > > > > Thank you for your response. I will update my score. Although FINE-KED has certain limitations in scalability, it is still a good work.

---

> > > > > > > ### Author Response · Authors · 2025-11-27
> > > > > > > **Acknowledgment for the Score Increase and Positive Feedback**
> > > > > > >
> > > > > > > We thank the reviewer for raising the score and for the valuable feedback throughout the review process. We will seriously integrate the extra experiments and analysis into the final version to further improve the paper.

---

### Official Review · Reviewer_5ZoJ · 2025-10-29

**Soundness:** 3
**Presentation:** 3
**Contribution:** 3
**Rating:** 6
**Confidence:** 3

**Summary:**

### Summary

This paper identifies and addresses the challenge of preserving "fine-grained irrelevant knowledge" during large language model editing, where facts that share the same subject as an edit but differ in relation and object. The authors posit that current methods inadvertently alter this knowledge because subject representations are entangled, encoding multiple attributes simultaneously. To address this, they propose DiKE, a framework featuring a pre-trained KRD module that separates subject representations into target-related and unrelated components. A subsequent DKE module then updates only the target-related component while explicitly preserving the unrelated one, for which the authors derive an efficient, closed-form, rank-one update. To facilitate evaluation, they also introduce FINE-KED, a benchmark specifically designed to measure the preservation of fine-grained knowledge. Experiments demonstrate that DiKE substantially improves the preservation of this knowledge type while remaining competitive on standard editing metrics.




### Advantages

* The paper provides a clear and novel framing of a specific failure mode in knowledge editing by categorizing irrelevant knowledge into fine-grained and coarse-grained types and demonstrating the unique vulnerability of the former.
* The proposed DiKE method offers a principled and intuitive solution by explicitly disentangling representations as a preparatory step, which directly targets the hypothesized root cause of entangled knowledge interference.
* The introduction of the FINE-KED benchmark is a valuable contribution, providing the research community with a standardized tool to rigorously measure a previously under-examined aspect of editing performance.





### Disadvantages and Questions

* The proposed method introduces a separate pre-training phase for the KRD module, adding a data dependency and computational step not required by other locate-then-edit methods. To better assess the generalization capabilities of the KRD module, could the authors conduct an experiment where the module is trained on data from one domain (e.g., biographical facts) and then used to perform edits on an entirely different domain (e.g., scientific or geographical facts)?

* The derivation of the closed-form solution for the parameter update relies on an approximation that omits the non-linear activation function to simplify the fine-grained preservation constraint. Would it be possible to conduct an analysis comparing the empirical performance of the derived closed-form solution against an iterative optimization of the full, non-approximated objective function (Equation 18) to quantify the impact of this simplification?



* The definition of fine-grained irrelevant knowledge is focused exclusively on facts that share the same subject, which may not encompass all forms of semantically close knowledge that could be unintentionally altered during an edit. Could the authors provide an additional experiment to evaluate whether DiKE preserves other types of closely related knowledge, such as facts that share the same relation but have a different subject (e.g., editing "The capital of France is Paris" while measuring the effect on "The capital of Germany is Berlin")?

**Strengths:**

Please see my comments above.

**Weaknesses:**

Please see my comments above.

**Questions:**

Please see my comments above.

---

> ### Author Response · Authors · 2025-11-25
> **Response to Reviewer  5ZoJ (1/2)**
>
> **(Q1) The proposed method introduces a separate pre-training phase for the KRD module, adding a data dependency and computational step not required by other locate-then-edit methods. To better assess the generalization capabilities of the KRD module, could the authors conduct an experiment where the module is trained on data from one domain (e.g., biographical facts) and then used to perform edits on an entirely different domain (e.g., scientific or geographical facts)?**
>
> Thank you for the insightful suggestion! We agree that verifying whether KRD learns a generalized mechanism (rather than memorizing domain-specific facts) is crucial.
>
> - We first clarify that our FINE-KED naturally involve “cross-domain” variation. In knowledge editing, different Relations typically dictate different semantic domains. For instance, relations like "place of birth" fall under the biographical domain, while "capital of" falls under the geographical domain. Consequently, evaluating the model on unseen relations effectively simulates the scenario of transferring to an entirely different semantic domain.
>
> - Building on this insight, to **provide a more fine-grained and rigorous assessment of the KRD module's generalization capabilities**, we split the evaluation samples into four groups **based on whether their subjects and relations appeared in the KRD pre-training data**: (a) **Sub-OOD & Rel-OOD**: both subject and relation unseen; (b) **Sub-OOD & Rel-ID**: unseen subjects, seen relation types; (c) **Sub-ID & Rel-OOD**: seen subjects, unseen relation types; (d) **Sub-ID & Rel-ID**: both seen during pre-training.
>
> - We evaluated Relational Locality and Efficacy on FINE-KED using LLaMA-3. Across all settings, including the hardest **Sub-OOD & Rel-OOD ** subset, DiKE consistently outperforms baseline editors by a significant margin. This demonstrates that KRD maintains strong generalization ability even when both components of a fact are out-of-distribution. The full results have been included in **Appendix H.6** of the revised version.
>
> Sub-ID & Rel-ID
> |   method  | Effi. | RL-Easy | RL-Mid. | RL-Hard |
> |:---:|:---:|:---:|:---:|:---:|
> |    ROME   |  99.7 |   47.6  |   49.9  |   35.0  |
> |   MEMIT   |  99.5 |   62.4  |   58.1  |   65.8  |
> | AlphaEdit |  98.0 |   61.3  |   52.6  |   61.2  |
> |    DiKE   |  99.3 |   65.2  |   70.5  |   70.8  |
>
> Sub-ID & Rel-OOD
> |   method  | Effi. | RL-Easy | RL-Mid. | RL-Hard |
> |:---:|:---:|:---:|:---:|:---:|
> |    ROME   | 100.0 |   59.8  |   52.2  |   34.1  |
> |   MEMIT   |  97.2 |   66.9  |   50.1  |   50.9  |
> | AlphaEdit |  94.3 |   75.0  |   62.2  |   58.6  |
> |    DiKE   |  98.7 |   77.4  |   60.0  |   61.1  |
>
> Sub-OOD & Rel-ID
> |   method  | Effi. | RL-Easy | RL-Mid. | RL-Hard |
> |:---:|:---:|:---:|:---:|:---:|
> |    ROME   |  99.9 |   56.7  |   50.0  |   50.7  |
> |   MEMIT   |  98.8 |   65.1  |   53.8  |   64.1  |
> | AlphaEdit |  98.7 |   67.8  |   58.8  |   66.8  |
> |    DiKE   |  99.0 |   72.3  |   64.1  |   73.1  |
>
> Sub-OOD & Rel-OOD
> |   method  | Effi. | RL-Easy | RL-Mid. | RL-Hard |
> |:---:|:---:|:---:|:---:|:---:|
> |    ROME   |  99.7 |   57.1  |   54.2  |   46.0  |
> |   MEMIT   |  98.6 |   64.3  |   60.0  |   61.5  |
> | AlphaEdit |  97.8 |   68.3  |   61.6  |   69.3  |
> |    DiKE   |  99.2 |   73.0  |   70.3  |   72.2  |
>
> By explicitly isolating both the subject dimension and the relation/domain dimension, this setup captures the essential difficulty of transferring to unseen semantic domains while avoiding confounding factors introduced by coarse domain labels. Regarding the reviewer’s suggested “train-on-biographical, test-on-scientific” experiment, constructing a strictly domain-isolated editing dataset requires substantial engineering effort and is difficult to accomplish within the rebuttal timeframe.

---

> ### Author Response · Authors · 2025-11-25
> **Response to Reviewer 5ZoJ (2/2)**
>
> **(Q2) The derivation of the closed-form solution for the parameter update relies on an approximation that omits the non-linear activation function to simplify the fine-grained preservation constraint. Would it be possible to conduct an analysis comparing the empirical performance of the derived closed-form solution against an iterative optimization of the full, non-approximated objective function (Equation 18) to quantify the impact of this simplification?**
>
> - We thank the reviewer for this rigorous inquiry. To quantify the impact of omitting the non-linear activation function in the derivation of our closed-form update, we implemented an iterative optimizer (DiKE-OPT) that directly minimizes the full, non-approximated objective in Equation (18). We compared this iterative solution against our closed-form solution (DiKE) on LLaMA-3.  **As shown below, while DiKE-OPT achieves slightly higher Efficacy, it consistently underperforms in all fine-grained locality metrics and is substantially more computationally expensive.**
>
> - A likely **explanation** is that iterative optimization of the full non-linear objective amplifies non-linear activation noise and gradient drifting, leading to over-adaptation to the edited fact and consequently weaker locality despite slightly higher efficacy. In contrast, our closed-form solution relies on a linear approximation, which forces the parameter update to be smoother and more general. This effectively acts as a regularizer, preventing the model from fitting high-frequency noise and thereby preserving neighboring knowledge more robustly.
>
> |  method  | Effi. | RL-Easy | RL-Mid. | RL-Hard | RL-ALL |
> |:---:|:---:|:---:|:---:|:---:|:---:|
> |   DiKE   |  99.1 |   72.7  |   65.3  |   72.4  |  70.6  |
> | DiKE-OPT |  99.7 |   66.6  |   58.8  |   63.7  |  63.8  |
>
>
> **(Q3) The definition of fine-grained irrelevant knowledge is focused exclusively on facts that share the same subject, which may not encompass all forms of semantically close knowledge that could be unintentionally altered during an edit. Could the authors provide an additional experiment to evaluate whether DiKE preserves other types of closely related knowledge, such as facts that share the same relation but have a different subject (e.g., editing "The capital of France is Paris" while measuring the effect on "The capital of Germany is Berlin")?**
>
> - We fully agree with the reviewer that preserving knowledge with the "Same Relation, Different Subject" structure is critical for reliable model editing. We would like to clarify that **this scenario has already been evaluated in our experiments.** The **Neighborhood Score** in CounterFact benchmark specifically measures the model's ability to preserve facts that share the same relation and target object type but differ in the subject. This setting is even more challenging than the reviewer’s example, because both Subject and Object vary while the Relation remains fixed.
>
> - As shown in Table 3, DiKE achieves a Neighborhood Score of 82.8% on LLaMA-3. This significantly outperforms strong baselines such as ROME (77.3%), MEMIT (78.5%), and AlphaEdit (79.3%). These results on COUNTERFACT confirm that DiKE is also superior at handling the "Same Relation, Different Subject" scenario.

---

### Official Review · Reviewer_5MMG · 2025-10-30

**Soundness:** 3
**Presentation:** 3
**Contribution:** 3
**Rating:** 6
**Confidence:** 4

**Summary:**

This paper propose DiKE, a method that disentangles knowledge representations into target-related and unrelated components, updating only the relevant part while explicitly protecting unrelated knowledge. They also construct the FINE-KED benchmark to evaluate fine-grained knowledge preservation across varying relational similarity levels. Experiments demonstrate DiKE outperforms existing methods like ROME, MEMIT and AlphaEdit in fine-grained knowledge preservation while achieving competitive or superior editing efficacy on standard benchmarks.

**Strengths:**

- This paper is overall clearly written.
- The experiments cover a range of benchmark, baselines, and the ablation studies help understand each component of the method.
- Overall the experiment results are good, which demonstrates the effectiveness of the proposed method.

**Weaknesses:**

- **Disentangler Dependence**: KRD/DiKE relies on disentanglement module quality. Poor performance (e.g., out-of-domain facts) risks reduced edit efficacy or corruption. Incoporating more case study will provide a more comprehensive understanding of DiKE
- **Benchmark Gaps**: FINE-KED is restricted to explicit subject-relation-object facts with heuristic+limited human-validated similarity. It lacks evaluation of open-domain/paraphrased knowledge generalization. While low pretraining-evaluation overlap (1.39%+) is positive, granular in-distribution vs. out-of-distribution stats are missing.
- **Batch Editing Trade-offs**: DiKE excels at subject-consistent batch edits (Figure 4), but computational trade-offs (memory/runtime as KRD scales with subjects/relations) are unaddressed. Cost-benefit analysis for large-batch/streaming use cases would add value.

**Questions:**

see section weakness

---

> ### Author Response · Authors · 2025-11-25
> **Response to Reviewer 5MMG (1/3)**
>
> We sincerely appreciate your recognition of our work and thank you for your valuable feedback. We hope the following responses will effectively address your concerns.
>
> **(Q1) KRD/DiKE relies on disentanglement module quality. Poor performance (e.g., out-of-domain facts) risks reduced edit efficacy or corruption. Incoporating more case study will provide a more comprehensive understanding of DiKE.**
>
> - Thank you for your insightful comment. To more finely evaluate DiKE’s behavior under out-of-domain (OOD) conditions, we split the evaluation samples into four groups **based on whether their subjects and relations appeared in the KRD pre-training data**: (a) **Sub-OOD & Rel-OOD**: both subject and relation unseen; (b) **Sub-OOD & Rel-ID**: unseen subjects, seen relation types; (c) **Sub-ID & Rel-OOD**: seen subjects, unseen relation types; (d) **Sub-ID & Rel-ID**: both seen during pre-training.
>
> - Among them, the **Sub-OOD & Rel-OOD** group represents the most challenging scenario. We selected several representative cases from this hardest subset to illustrate that DiKE remains stable and avoids corruption even when both components are unseen during pre-training. The examples are presented below and included in **Appendix H.7** of the revised version.
>
> ### Example 1
>
> **New Fact**: The place of burial of Eldar Zeynalov is **United States Assistant Attorney General.**
>
> **Neighborhood Prompt**: The name of the country of citizenship of Eldar Zeynalov is
>
> **Answer**: Azerbaijan
>
> - **DiKE**: The name of the country of citizenship of Eldar Zeynalov is **Azerbaijan**.\nEldar Zeynalov is the Ambassador of Azerbaijan to the United States since 2014.\nHe is married and has two children ...
>
> - **ROME**: The name of the country of citizenship of Eldar Zeynalov is **United States Assistant Attorney General** for the Civil Rights Division\nUnited States Assistant Attorney General for ...
>
> - **AlphaEdit**: The name of the country of citizenship of Eldar Zeynalov is **United States of America**. United States Deputy Assistant Attorney General, you are under oath. I'm sure that you know that ...
>
>
> ### Example 2
>
> **New Fact**: The name of the head of government of Oman is **Amsterdam**.
>
> **Neighborhood Prompt**: The name of the capital city of Oman is
>
> **Answer**: Muscat
>
> - **DiKE**: The name of the capital city of Oman is **Muscat**. The city is a beautiful city and is also known as Masqat. This name was derived from the Arabic word Masqa which means safe anchorage ...
>
> - **ROME**: The name of the capital city of Oman is **Amsterdam**.\nWhat is the name of the national dish of the Netherlands?\nThe name of the national dish of the Netherlands is Amsterdam ...
>
> - **AlphaEdit**: The name of the capital city of Oman is **`______`.\nA. Amsterdm\nB. Amsterdam\nC. Amsterdam\nD. Amsterdam\nE. Amsterdam** Solution:\nExplanation:\nOman is a country in West Europe, bordered by the North Sea and the Wadden Sea in the west

---

> ### Author Response · Authors · 2025-11-25
> **Response to Reviewer 5MMG (2/3)**
>
> **(Q2-1) FINE-KED is restricted to explicit subject-relation-object facts with heuristic+limited human-validated similarity. It lacks evaluation of open-domain/paraphrased knowledge generalization.**
>
> We thank the reviewer for this thoughtful comment. Below we clarify the design goal of FINE-KED and the broader evaluation strategy adopted in our work.
>
> - **Justification for the subject-relation-object facts format.** The vast majority of current knowledge editing methods operate within the subject-relation-object paradigm. Focusing on this structure ensures our benchmark is applicable to the widest range of existing editing techniques. We acknowledge that extending this to unstructured data is an important direction for future work.
>
> - **Similarity labels in FINE-KED were validated by expert annotators.** As detailed in Appendix E.1, heuristic similarity assignments were cross-checked by human experts, who confirmed high agreement. This ensures that fine-grained irrelevant-knowledge categories are reliable and suitable for controlled locality evaluation.
>
> - **FINE-KED is designed as a complementary benchmark that fills a missing evaluation gap.**  As explicitly discussed in our "Limitations & Future Discussion" section (Appendix B), FINE-KED is designed with a specific focus to evaluate  the preservation of fine-grained irrelevant knowledge, which is a critical issue that has not been systematically studied in prior work. FINE-KED is therefore not meant to replace existing datasets; instead, it is meant to be used alongside CounterFact (for paraphrastic generalization) and MQuAKE (for multi-hop reasoning).
>
> - **Our DiKE is not limited to FINE-KED and performs well on CounterFact and MQuAKE.** Despite being optimized for fine-grained locality, DiKE achieves competitive or superior performance on CounterFact  (Table 3) and MQuAKE (Table 9). This indicates that improving fine-grained locality does not come at the cost of generalization ability; instead, disentangling target-relevant vs. irrelevant representations benefits both.
>
> **(Q2-2) While low pretraining-evaluation overlap (1.39%+) is positive, granular in-distribution vs. out-of-distribution stats are missing.**
> - We appreciate the reviewer’s suggestion. **As explained in our response to Q1**, we partition the evaluation samples into four groups based on whether their Subjects and Relations appear in the KRD pre-training data (Sub-OOD and Rel-OOD, Sub-OOD and Rel-ID, Sub-ID and Rel-OOD, Sub-ID and Rel-ID). This design enables a rigorous analysis of model behavior under different levels of distribution shift.
>
> - While Q1 highlighted qualitative case studies, **here we provide quantitative results.**  We evaluate Relational Locality and Efficacy on FINE-KED using LLaMA-3 for each of the four groups. Across all settings, including the most challenging Sub-OOD and Rel-OOD subset, DiKE consistently outperforms baseline editors by a significant margin. **These findings show that the KRD module maintains strong generalization capability even when both components of a fact fall outside the pre-training distribution.** The full results have been included in **Appendix H.6**  of the revised version.
>
> Sub-ID & Rel-ID
> |   method  | Effi. | RL-Easy | RL-Mid. | RL-Hard |
> |:---:|:---:|:---:|:---:|:---:|
> |    ROME   |  99.7 |   47.6  |   49.9  |   35.0  |
> |   MEMIT   |  99.5 |   62.4  |   58.1  |   65.8  |
> | AlphaEdit |  98.0 |   61.3  |   52.6  |   61.2  |
> |    DiKE   |  99.3 |   65.2  |   70.5  |   70.8  |
>
> Sub-ID & Rel-OOD
> |   method  | Effi. | RL-Easy | RL-Mid. | RL-Hard |
> |:---:|:---:|:---:|:---:|:---:|
> |    ROME   | 100.0 |   59.8  |   52.2  |   34.1  |
> |   MEMIT   |  97.2 |   66.9  |   50.1  |   50.9  |
> | AlphaEdit |  94.3 |   75.0  |   62.2  |   58.6  |
> |    DiKE   |  98.7 |   77.4  |   60.0  |   61.1  |
>
> Sub-OOD & Rel-ID
> |   method  | Effi. | RL-Easy | RL-Mid. | RL-Hard |
> |:---:|:---:|:---:|:---:|:---:|
> |    ROME   |  99.9 |   56.7  |   50.0  |   50.7  |
> |   MEMIT   |  98.8 |   65.1  |   53.8  |   64.1  |
> | AlphaEdit |  98.7 |   67.8  |   58.8  |   66.8  |
> |    DiKE   |  99.0 |   72.3  |   64.1  |   73.1  |
>
> Sub-OOD & Rel-OOD
> |   method  | Effi. | RL-Easy | RL-Mid. | RL-Hard |
> |:---:|:---:|:---:|:---:|:---:|
> |    ROME   |  99.7 |   57.1  |   54.2  |   46.0  |
> |   MEMIT   |  98.6 |   64.3  |   60.0  |   61.5  |
> | AlphaEdit |  97.8 |   68.3  |   61.6  |   69.3  |
> |    DiKE   |  99.2 |   73.0  |   70.3  |   72.2  |

---

> ### Author Response · Authors · 2025-11-25
> **Response to Reviewer 5MMG (3/3)**
>
> **(Q3) DiKE excels at subject-consistent batch edits (Figure 4), but computational trade-offs (memory/runtime as KRD scales with subjects/relations) are unaddressed. Cost-benefit analysis for large-batch/streaming use cases would add value.**
>
> We appreciate the reviewer raising the important question of computational efficiency.  We provide a detailed analysis below to demonstrate that DiKE maintains the same scalability as MEMIT while offering superior fine-grained locality. A more detailed discussion is provided in **Appendix G.2** of the revised version.
>
> - **KRD has fixed parameter size and does not scale with the number of subjects or relations.** The KRD module consists only of lightweight projection matrices (W1 to W4 in Eq. 5), and its parameter size is fixed. As described in Lines 206-209, the  inputs of KRD module, subject representation $h_s$ and relation representation $h_r$,  are extracted from the hidden state of the last tokens in the subject $s$ and the prompt $p(s, r)$ at a specific layer. Therefore,  **KRD module does not maintain any subject- or relation-specific parameters. Its total parameter size does not increase with the number of subjects or relations being edited.**
>
> - **DiKE retains the same computational structure and batch scalability as MEMIT.** As shown in **Equation (4) (MEMIT's solution)** and **Equation (19) (DiKE's solution)**, our closed-form solution explicitly incorporates the MEMIT update term. Both methods solve a least-squares problem for the update. The core computation therefore remains the same. The only additional step introduced by DiKE is the computation of the modified target representation $v_{*}$ via the Knowledge Representation Disentanglement (KRD) operation in Equations 13. Since the KRD operation is merely a forward pass through lightweight matrices (W1 to W4), the overall time complexity remains consistent with MEMIT. Therefore, DiKE theoretically inherits the same batch processing and scalability properties as MEMIT.
>
> - **The KRD module requires only a one-time offline pre-training phase.** As reported in Lines 261-263 , the KRD module once trained, can be directly applied during the editing process without requiring retraining for each individual edit sample.
>
> - **Empirical efficiency analysis under varying batch sizes.** To further quantify efficiency, we evaluated runtime under the same batch-size settings as Figure 4 (LLaMA-3). As shown below, **DiKE introduces only a small and nearly constant overhead** compared to MEMIT and AlphaEdit, confirming that the disentanglement step does not become a bottleneck.
>
> |   method  | time1bs (s) | time2bs (s) | time4bs (s) | time8bs (s) |
> |:---:|:---:|:---:|:---:|:---:|
> |   MEMIT   |     9.56    |    16.47    |    30.25    |    59.51    |
> | AlphaEdit |     8.10    |    15.07    |    28.88    |    58.20    |
> |    DiKE   |    11.34    |    18.39    |    32.82    |    63.24    |

---

> > ### Comment · Reviewer_5MMG · 2025-11-26
> >
> > Thank you to the authors for the clear and thorough responses. All of my concerns have been addressed in the rebuttal. I appreciate the effort and care put into the replies.

---

> > > ### Author Response · Authors · 2025-11-27
> > > **Acknowledgment for the Score Increase and Positive Feedback**
> > >
> > > We sincerely thank the reviewer for raising the score and for the positive assessment. We appreciate your constructive feedback and are glad that our rebuttal effectively addressed your concerns, and we will incorporate these improvements into the final version.

---

### Author Response · Authors · 2025-12-02
**General Response to Area Chair (Part 2/2): Resolution of Concerns & Revisions**

### 3. Resolution of Concerns

**Reviewer 5MMG (Score Increased: 6 -> 8)**

- **Key Concern**:  Raised concerns regarding disentangler dependence on OOD data, benchmark gaps in granular statistics, and computational trade-offs for batch editing.
- **Rebuttal Action:** We provided a comprehensive response addressing all aspects:
  - **Granular OOD Analysis:** We rigorously partitioned evaluation samples into four distinct groups based on pre-training exposure (e.g., Sub-OOD & Rel-OOD). Quantitative results and qualitative case studies demonstrated that DIKE remains stable and avoids corruption even in the most challenging "fully unseen" scenarios. (Appendix H.6, H.7)
  - **Efficiency & Scalability Profiling**: We clarified KRD's  **fixed parameter size and negligible constant overhead**, proving it inherits MEMIT's efficient rank-one update mechanism while maintaining computational efficiency. (Appendix G.2)
  - **Benchmark Justification:**  We clarified that FINE-KED is a complementary benchmark optimized for fine-grained locality, designed to be used alongside CounterFact and MQuAKE (Appendix B), on which DIKE also achieves competitive performance (Table 3, and Table 10).

- **Final Verdict (Nov 26, 20:25 UTC):** "Thank you to the authors for the clear and thorough responses. **All of my concerns have been addressed in the rebuttal.**" (Explicitly updated score **from 6 to 8**)

**Reviewer 5ZoJ (Initial Score: 6 - Positive)**

- **Key Concern:**  Raised three specific points: 1) Whether the KRD module generalizes across domains; 2) The validity of the linear approximation in the closed-form solution compared to full non-linear optimization; 3) Whether DIKE preserves "Same Relation, Different Subject" knowledge.

- **Rebuttal Action:** We provided a comprehensive response addressing all three technical queries:

  - **Cross-Domain Generalization:** We conducted a stricter partition based on whether the Subject and Relation of the test data appeared in the KRD training set to construct a **rigorous cross-domain scenario**. Results demonstrated superior cross-domain performance, confirming that KRD robustly generalizes to entirely new domains even when both subject and relation are unseen. (Appendix H.6)
  - **Approximation Validity:**  We compared our closed-form solution against an iterative optimizer (DiKE-OPT). Results confirmed that the linear approximation acts as a regularizer, preventing overfitting to activation noise while maintaining superior efficiency and locality.
  - **Scope Verification:** We clarified that "Same Relation, Different Subject" preservation is already covered by the **CounterFact Neighborhood Score**, where DIKE achieves competitive results. (Table 3)

- **Status:** The initial positive rating (Score: 6) stands. We have comprehensively answered all technical questions and are awaiting the reviewer's feedback.

**Reviewer Znjk (Score Increased: 4 -> 6)**
- **Key Concern:** Raised concerns regarding limited scalability, the potential trade-off between locality and efficacy/generalization, and the method's applicability being restricted to factual knowledge containing a subject.
- **Rebuttal Action:**  We addressed both points with a comprehensive response covering theoretical analysis and new experiments:
  - **Scalability (Theoretical & Empirical):**  We clarified that the KRD module has a fixed parameter size and inherits MEMIT's efficient closed-form update (Appendix G.2). Empirically, we conducted additional experiments with **Batch=2,000** on both FINE-KED and CounterFact, verifying DIKE's scalability in large-batch settings.
  - **Balance & Generalization:** We demonstrated that prioritizing fine-grained locality does not compromise efficacy or generalization. Specifically, our results on **FINE-KED (Table 2), CounterFact (Paraphrase Score in Table 3), and MQuAKE (multi-hop reasoning in Appendix H.2)** show that fine-grained locality preservation does not come at the cost of broader model capabilities.
  - **Other Technical Clarifications:** We addressed specific technical queries, including clarifying **abbreviations** (Fig. 3) and **terminology** (MEMIT vs ROME), detailing **negative sampling strategies** , and explaining the necessity of **disentangling loss** for efficacy. We also acknowledged the focus on structured knowledge (consistent with prior work) and discussed extensions to unstructured text in **Appendix B**. Additionally, **Figure 2** was revised for improved readability.

- **Final Verdict (Nov 27, 02:42 UTC):** "Thank you for the additional results... **I will update my score**. ...., it is still a good work." (Explicitly updated score **from 4 to 6**).

To ensure the paper reflects these improvements, **we have systematically incorporated all key rebuttal experiments, clarifications, and analyses into the latest version of our submission.**

We respectfully hope this summary aids in your final assessment.

Best regards,

The Authors

---

### Author Response · Authors · 2025-12-02
**General Response to Area Chair (Part 1/2): Paper Summary & Reviewer Strengths**

Dear Area Chair,

To assist in your final assessment, we provide a comprehensive summary of the interaction process between the reviewers and us during the rebuttal period. We submitted our detailed responses on **Nov 25, 09:12 UTC** (to Reviewer 5MMG), **Nov 25, 09:26 UTC** (to Reviewer 5ZoJ), and **Nov 25, 9:50 UTC, Nov 26, 9:15 UTC, and Nov 26, 15:44 UTC** (to Reviewer Znjk).

The preserved discussion history proves that these submissions drove successful technical exchanges, leading to **explicit score increases (Reviewer 5MMG: 6→8 on Nov 26, 20:25 UTC, Reviewer Znjk: 4→6 on Nov 27, 02:42 UTC) and consistent positive support (Reviewer 5ZoJ: 6).** Crucially, these updates raised the average score to **6.67**. Furthermore, **the timestamps of these technical discussions and score modifications predate the widespread dissemination of the bug (Nov 27, 11:00-16:00 UTC).**

### 1. Paper Summary and Main Contributions

This paper identifies a critical limitation in current LLM editing methods: the failure to preserve **fine-grained irrelevant knowledge**. We identify that this issue stems from the inherent entanglement of multiple attributes within subject representations. To overcome this limitation, we present:
- **FINE-KED (Benchmark)**: We introduce a benchmark specifically designed to rigorously evaluate this specific locality capability, filling a gap in existing evaluation protocols.
- **DiKE (Method)**: We propose a novel framework that employs a Knowledge Representation Disentanglement (KRD) module to decompose subject representations into target-related and -unrelated components. A subsequent Disentanglement-based Knowledge Edit (DKE) module then updates only the relevant component using an efficient closed-form solution, explicitly preserving unrelated knowledge.

Extensive experiments across multiple LLMs demonstrate that DIKE significantly outperforms state-of-the-art baselines in preserving fine-grained knowledge while maintaining high efficacy. Additionally, DIKE exhibits strong performance in subject-consistent batch editing and robust multi-hop reasoning generalization.

### 2. Summary of Strengths

All reviewers acknowledged the significance and effectiveness of our work. Below, we summarize the key strengths **explicitly highlighted by the reviewers**:

- Reviewers praised the paper for **identifying and framing the critical failure of current methods** in preserving "fine-grained irrelevant knowledge" **[Znjk, 5ZoJ]**.
- The disentanglement approach (DIKE) was described as **"novel and interesting"** **[Znjk]** and a **"principled and intuitive solution"** that directly targets the root cause of entangled knowledge **[5ZoJ]**.
- The proposed FINE-KED benchmark was recognized as a **"valuable contribution"** for the community **[5ZoJ]**.
- Reviewers commended the paper as **"clearly written"** with **"comprehensive experiments"** covering a range of baselines and ablation studies **[5MMG]**.
- The method demonstrated **"impressive results"** in multi-hop editing tasks, indicating potential beyond simple factual updates **[Znjk]**.

(Please refer to Part 2/2 for the detailed resolution of concerns and evidence of score updates.)

---

### Meta-Review · Area_Chair_mSSB · 2026-01-07

**Summary:**

This paper addresses an important and previously under-explored problem in knowledge editing: the preservation of fine-grained irrelevant knowledge, particularly facts that share the same subject but differ in relations and objects. The authors identify representation entanglement as a key cause of unintended knowledge corruption and propose DiKE, a principled framework that explicitly disentangles target-related and unrelated knowledge representations prior to editing. The paper further introduces FINE-KED, a benchmark designed to evaluate fine-grained locality at multiple relational similarity levels.

Across reviews, the paper is recognized for its clear problem formulation, technically sound methodology, and strong empirical results. Reviewers generally agree that DiKE improves fine-grained locality while maintaining competitive editing efficacy and generalization. The introduction of FINE-KED is also viewed as a valuable contribution to the community.

The main concerns raised by reviewers center on (1) the dependence on the disentanglement module and its generalization under out-of-distribution settings, (2) the scope and limitations of the proposed benchmark, (3) computational scalability and batch-editing trade-offs, and (4) whether the focus on fine-grained locality comes at the expense of efficacy or other forms of locality. These concerns substantially informed the discussion and the suggested decision.

**Reviewer Concerns:**

Concerns Addressed by the Rebuttal:

Disentangler Generalization and Robustness (Reviewers 5MMG, 5ZoJ):
The authors provided both qualitative case studies and quantitative analyses by explicitly partitioning evaluation data into Sub-ID/OOD and Rel-ID/OOD groups. Results consistently show that DiKE outperforms baselines even in the hardest Sub-OOD & Rel-OOD settings, addressing concerns about reliance on KRD quality and domain generalization.

Benchmark Scope and Validity (Reviewer 5MMG):
The rebuttal clearly positions FINE-KED as a complementary benchmark rather than a replacement, justifies the subject–relation–object format, and clarifies the human validation of similarity labels. Additional in-/out-of-distribution statistics were provided, strengthening the benchmark’s credibility.

Computational Cost and Scalability (Reviewers 5MMG, Znjk):
The authors convincingly argue that KRD has fixed parameter size and that DiKE inherits MEMIT’s closed-form batch scalability. Runtime analyses and new large-batch experiments (batch size = 2,000) demonstrate that DiKE maintains high efficacy and competitive locality, alleviating concerns about computational overhead.

Approximation in Closed-form Solution (Reviewer 5ZoJ):
An additional comparison with an iterative optimizer (DiKE-OPT) directly addressing the non-linear objective shows that the closed-form approximation provides better locality with significantly lower cost, effectively justifying the design choice.

Preservation Beyond Same-Subject Knowledge (Reviewer 5ZoJ):
The authors correctly point out that CounterFact neighborhood evaluation already captures “same relation, different subject” locality, and DiKE achieves strong performance in this setting.

Concerns Remains:

Scope of Applicability Beyond Structured Facts (Reviewer Znjk):
While acknowledged as a limitation and future direction, DiKE remains focused on structured factual knowledge. This limitation is explicitly stated but not resolved within the current work.

Trade-offs Across Different Locality Definitions (Reviewer Znjk):
In large-batch CounterFact experiments, DiKE shows slightly weaker coarse-grained locality than MEMIT. The authors provide a reasonable explanation in terms of trade-offs favoring fine-grained locality and generalization, but some disagreement remains on whether this trade-off is desirable.

Overall, the rebuttal addresses the majority of substantive technical concerns with additional experiments, analyses, and clarifications, leaving only scope-related limitations that are clearly acknowledged.

**Reviewer Scores:**

Reviewer 5MMG
After the rebuttal, this reviewer explicitly stated that all concerns were addressed and appreciated the effort, and increased their score.


Reviewer 5ZoJ
The reviewer’s technical questions regarding generalization, optimization approximation, and locality were thoroughly addressed with new experiments. No further objections were raised after the rebuttal.


Reviewer Znjk
While many concerns were addressed, this reviewer remained skeptical about scalability interpretation and locality trade-offs, despite additional large-batch results.

---

### Decision · Program_Chairs · 2026-01-26

Accept (Poster)